# Metabolomics Study on Pathogenic and Non-pathogenic *E. coli* with Closely Related Genomes with a Focus on Yersiniabactin and Its Known and Novel Derivatives

**DOI:** 10.3390/metabo10060221

**Published:** 2020-05-28

**Authors:** Mareike Schulz, Vasiliki Gaitanoglou, Olena Mantel, Yannick Hövelmann, Florian Hübner, Ulrich Dobrindt, Hans-Ulrich Humpf

**Affiliations:** 1Institute of Food Chemistry, Westfälische Wilhelms-Universität Münster, Corrensstraße 45, 48149 Münster, Germany; m_sch065@uni-muenster.de (M.S.); vasilikigaitanoglou@yahoo.de (V.G.); yannick.hoevelmann@uni-muenster.de (Y.H); florian.huebner@uni-muenster.de (F.H.); 2Institute of Hygiene, Universitätsklinikum Münster, Mendelstraße 7, 48149 Münster, Germany; olena.mantel@ukmuenster.de

**Keywords:** metabolomics, bacteria, *Escherichia coli*, Nissle 1917, 83972, CFT073, yersiniabactin, escherichelin, ulbactin B, novel derivatives

## Abstract

The *Escherichia coli* (*E. coli*) strains Nissle 1917 (EcN), 83972 and CFT073 are closely related but differ in their phenotypes and pathogenicity. The aim of this study was to compare the metabolome of these strains based on metabolomic data analysis of bacterial samples using liquid chromatography-high resolution mass spectrometry (LC-HRMS). The strains were cultivated in minimum essential medium at 37 °C for 6 h. The sterilized culture supernatant was analyzed, followed by data processing to create feature lists, and statistical analysis to identify discriminating features in the metabolomes of the three strains. Metabolites were identified using the exact masses, isotope patterns, and fragmentation spectra. The results showed that the metabolome of EcN differs significantly from the metabolomes of *E. coli* 83972 and CFT073. Based on the analysis, yersiniabactin (Ybt), its metal complexes, and its known structural derivatives escherichelin and ulbactin B were identified as discriminating features; the latter has not been described for *E. coli* before. Additionally, novel Ytb derivatives were found and tentatively identified by LC-MS/HRMS. All these metabolites were determined in significantly higher levels in the metabolome of EcN compared to *E. coli* 83972, which may explain a large part of the observed differences of the metabolomes.

## 1. Introduction

*Escherichia coli* (*E. coli*) belongs to the family of Enterobacteriaceae and mainly forms a symbiosis with their host. However, besides these commensal strains, some strains are pathogens and able to cause serious diseases. Therefore, *E. coli* strains can be subdivided according to their pathogenic potential and lifestyle into commensal *E. coli*, intestinal pathogenic *E. coli* (IPEC), and extraintestinal pathogenic *E. coli* (ExPEC) [1,2]. The latter group includes uropathogenic *E. coli* (UPEC) strains, which cause approximately 80% of uncomplicated symptomatic urinary tract infections (UTIs) [2,3,4]. The most common form of UTI is asymptomatic bacteriuria (ABU), which involves bacterial colonization of the bladder without causing severe symptoms of UTI. The ABU isolate *E. coli* 83972 is the best-characterized ABU isolate to date, which can efficiently colonize the human bladder in high numbers without causing overt symptoms of UTI [5,6]. In contrary, the highly virulent UPEC strain CFT073 has been isolated from a urosepsis case [7]. The fecal non-pathogenic *E. coli* isolate Nissle 1917 (EcN) is used as a probiotic agent against chronic inflammatory bowel diseases [8,9].

The uptake of iron from the environment is important for bacterial growth. During UTI, the bacteria grow in urine inside the bladder, where the supply of free iron is very limited. Similarly, other transition metal ions, e.g., Ni(II), Mn(II), and Cu(II), are also important growth factors required for efficient growth under anaerobic conditions or protection against oxidative stress [10]. Therefore, expressing effective metal uptake systems is important for bacterial survival and growth during infection. One type of iron acquisition systems is represented by siderophores, which are a group of chemically different secondary metabolites [4,11,12]. The three *E. coli* strains EcN, 83972, and CFT073 are able to produce the siderophores enterobactin, aerobactin, and salmochelin. However, only EcN and *E. coli* strain 83972 have the ability to produce yersiniabactin (Ybt), the structure of which is shown in Figure 1 [12,13,14,15]. *E. coli* strain CFT073, on the other hand, carries the determinant required for the biosynthesis of Ybt, but due to mutations in several open reading frames (ORFs), this strain is not able to produce this siderophore [16].

Ybt is much discussed in relation to UTIs. This metallophore got its name from *Yersinia pestis*, which also belongs to the Enterobacteriaceae. The high pathogenicity island (HPI), which encodes the proteins required for the biosynthetic pathway as well as the regulation and transport of Ybt, has been found in various species of the family of Enterobacteriaceae [17,18,19]. Besides its property as a chelating agent for Fe(III), Ybt also shows the ability to complex additional metal ions, for example, Ga(III), Al(III), Ni(II), Co(III), and Cr(III). The complexation with Cu(II) was observed during UTI as well and confers a protective effect of the bacteria against copper intoxication [18,20,21,22]. The biosynthetic pathway of Ybt was described in vitro by Miller et al. [23]. However, not many intermediates of the biosynthetic pathway have been analyzed yet. One example is escherichelin (Figure 1), which has recently been identified and detected during UTI caused by *E. coli*. It was previously known as HPTT-COOH (hydroxyphenyl-thiazolyl-thiazolinyl-carboxylic acid) and described as a synthetic inhibitor of pyochelin, a siderophore from *Pseudomonas aeruginosa*, which is structurally similar to Ybt [23,24,25,26,27,28,29]. Other structurally diverse Ybt derivatives can be produced by different bacterial strains [30]. Ulbactin B (Figure 1), which is one of several different ulbactins, is a derivative isolated from marine bacteria [31] and has not yet been described in *E. coli*.

Genetically, *E. coli* strains EcN, 83972 and CFT073, are closely related. They can be allocated to the phylogenetic lineage B2, and they belong to the same sequence type ST 73. Their different pathogenic potential might result from mutations in shared genomic regions and from differential gene expression rather than from differences in their genome content. A comparison of the genome content showed that EcN is more closely related to *E. coli* CFT073 than to *E. coli* 83972 [8,32,33,34,35,36]. Whereas UPEC strain CFT073 (serotype O6:K2:H1) expresses several virulence factors, like the toxin α-hemolysin, which is able to damage eukaryotic cells, and type 1, P, and F1C fimbrial adhesins [2,7,33], probiotic EcN (serotype O6:K5:H1) can express type 1 and F1C fimbriae, as well as curli fimbriae [37]. This strain is also able to produce two antimicrobial peptides, microcin M and H47 [38], but it lacks the ability to produce toxins [37]. Both strains, *E. coli* CFT073 and EcN, express flagella and are motile to varying degrees [2,8,37]. The asymptomatic bacteriuria isolate *E. coli* 83972 (serotype OR:K5:H2 (R, rough)) exhibits the weakest motility phenotype of all three strains. In addition, the expression of functional type 1 and P fimbriae as well as of α-hemolysin has been lost [2,6,39,40,41,42,43]. In regard to biofilm formation, which is also widely associated with infections, *E. coli* 83972 shows the best formation when comparing the three *E. coli* strains growing in different media, while EcN shows significantly higher biofilm formation than *E. coli* CFT073 [44].

These three strains have already been extensively compared on the genetic level because of their different phenotypes. However, so far there are hardly any metabolomic studies available on the differences of the small extracellular metabolites of these *E. coli* strains. A recently published study compared the metabolome of EcN and some other *E. coli* strains from the phylogroup B2, which are closely related to EcN and included CFT073 and 83972, and found differences in the metabolite profile of arginine biosynthesis-related metabolites [45].

The present study focused on small extracellular metabolites (exometabolome) in the sterile supernatant of cultures of the *E. coli* strains EcN, 83972, and CFT073. Therefore, the metabolome of these strains was compared by the application of non-targeted metabolomics based on reversed phase liquid chromatography high-resolution mass spectrometry (RP-LC-HRMS). The aim was to identify differences in their metabolomes, which might contribute to the different phenotypes of the strains.

## 2. Results

### 2.1. Metabolomics Study of the E. coli Strains EcN, 83972, and CFT073

The metabolome of the closely related *E. coli* strains EcN, 83972, and CFT073 cultivated in minimum essential medium (MEM) for 6 h was compared by the application of metabolomics based on LC-HRMS. Every strain was cultivated in three biological replicates and the samples were each analyzed in triplicate by LC-HRMS. Feature lists of the analyzed samples of the *E. coli* strains and the culture medium used as a blank sample were obtained using the data processing tool MZmine 2 [46]. Afterwards, they were compared using multivariate and univariate statistical analysis to find discriminating features, depending on the *m*/*z*, the retention time, and the intensities, between the *E. coli* strains.

The principal component analysis (PCA) scores plot of the multivariate statistical analysis in Figure 2 showed significant differences of the metabolome of EcN compared to the metabolome of the *E. coli* strains 83972 and CFT073 and the blank sample (represented by the different colors). This difference was due to the principal component 1, which accounts for 24.2% of the variance in the datasets. With regard to principal component 2 (14.4%), the metabolome of *E. coli* CFT073 also seemed to be distinguishable from the metabolome of the other strains. It displayed smaller differences to the metabolome of *E. coli* 83972 than to the metabolome of EcN. The features derived from the metabolome of *E. coli* 83972 exhibited the least difference in comparison to those of the blank sample.

The discriminating features responsible for this significant difference of the metabolome of EcN were determined by evaluating the corresponding PCA loadings plot (Figure 2). The red marked area in the plot marks the distinctive features, which were numbered from *1 to *15 and are reported in Table 1. These features were mainly responsible for the difference in relation to the principal component 1. The difference with respect to principal component 2 can be explained, for example, by feature *16, which is discussed below. Since the metabolome of EcN differed more from the other two, a comparison was carried out by means of univariate statistical analysis between the *E. coli* strains EcN vs. CFT073 as well as between EcN vs. 83972 for a better visualization of the discriminating features. The resulting volcano plots are shown in Figure 3 using the same numbering of the discriminating features as the PCA loadings plot (Figure 2).

Comparing the two volcano plots, the features *1 to *15 (Table 1) are some of those exhibiting a clear distinction between the metabolome of EcN and the other two *E. coli* strains. Feature *16, on the other hand, showed a more pronounced difference when comparing the metabolomes of *E. coli* strains EcN and CFT073, which was much less apparent between EcN and 83972. However, when comparing the metabolome of *E. coli* strains EcN and CFT073, the metabolites corresponding to the features (*1 to *15) showed greater differences in their expression levels and greater statistical significance than when comparing the metabolomes of EcN and 83972. Thus, these selected features may have a bigger impact when comparing EcN against *E. coli* CFT073. Nevertheless, these features also seem to distinguish the metabolome of EcN from that of *E. coli* 83972. When comparing the intensities of the different discriminating features, some of these characteristics were also detected in the metabolome of *E. coli* 83972. However, these features showed significantly lower intensities when compared to EcN, so that the metabolome of EcN showed a significant difference in the PCA scores plot. The intensities of the chosen features that were obtained by processing the recorded HRMS data with MZmine 2 [46] are presented in the Appendix A.

All 16 features were investigated based on their exact mass, isotope pattern, and fragmentation behavior. It was noticeable that all features *1 to *15, with the exception of *16, showed a characteristic isotope pattern, which indicated the presence of sulfur. Additionally, some features showed a characteristic isotope pattern, indicating the occurrence of iron and copper (Appendix A). Based on fragmentation experiments, it was possible, besides the prediction of molecular formulas, to assign some features to a common core structure, which was apparently cleaved by in-source fragmentation. Thus, some features could be grouped, which reduced the number of possible metabolites responsible for the significant difference of the metabolome of EcN. Table 1 lists the features that represent the [M+H]^+^ ions for the most significant metabolites as well as some major fragments formed by in-source fragmentation. The respective numbers correspond to the markings in the PCA loadings plot and the volcano plots (Figure 2 and Figure 3). Some of these features have been identified as already known metabolites while others are still so far unknown derivatives. For a better overview, features are labeled with an uppercase asterisk and ascending numbers. Newly detected compounds are labeled as bold numbers and isomeric forms are differentiated by capital letters for those showing differences in the retention time and fragmentation pattern. Isomers, which differ in retention time but not in fragmentation pattern, are indicated by the same capital letter with different subscript numbering, e.g., **1**-A and **1**-B are different isomers with distinct fragmentation patterns and retention times while Ybt-A_1_ and Ybt-A_2_ show the same fragmentation pattern but a different chromatographic behavior.

By means of the fragmentation pattern, the characteristic isotope pattern, which indicated the presence of sulfur, and the predicted molecular formulas, the known siderophore yersiniabactin (Ybt, features No. *1 and *3, [M+H]^+^: C_21_H_28_N_3_O_4_S_3_^+^, calculated: *m*/*z* 482.1236, measured: *m*/*z* 482.1229, Δm −1.5 ppm), which is known to consist of two diastereomers [18], was identified by LC-MS/HRMS (MS^2^) and LC-MS/MS/HRMS (MS^3^). Additionally, for the features showing a characteristic isotope pattern (Appendix A) indicating the occurrence of iron and copper, the known metal complexes of Ybt with iron (Fe(III)-Ybt, feature No. *5, [M+H]^+^: C_21_H_25_FeN_3_O_4_S_3_^+^, calculated: *m*/*z* 535.0351, measured: *m*/*z* 535.0340, Δm −2.1 ppm) and copper (Cu(II)-Ybt, feature No. *14, [M+H]^+^: C_21_H_26_CuN_3_O_4_S_3_^+^, calculated: *m*/*z* 543.0376, measured: *m*/*z* 543.0369, Δm −1.3 ppm) were also confirmed by MS^2^ and MS^3^ data and a comparison to the literature [18,20].

These metabolites showed the same neutral loss (NL) of 187 Da during in-source fragmentation, which has also been observed for several metal complexes of Ybt [20]. Additionally, two unknown metabolites with *m*/*z* 498.1178 (feature No. *6, [M+H]^+^: C_21_H_28_N_3_O_5_S_3_^+^, calculated: *m*/*z* 498.1186, measured: *m*/*z* 498.1178, Δm −1.5 ppm, abbreviation: **1**-A) and 601.1270 (feature No. *8, [M+H]^+^: C_24_H_33_N_4_O_6_S_4_^+^, calculated: *m*/*z* 601.1277, measured: *m*/*z* 601.1270, Δm −1.2 ppm, abbreviation: **2**-A) for the [M+H]^+^ ion showing the same NL of 187 Da were detected. Further investigations by MS^2^ and MS^3^ experiments showed that these could be novel derivatives of Ybt (see detailed MS studies in Section 2.4). During the investigations, more of these novel unknown derivatives were found and are described below (see Section 2.4.). Furthermore, one metabolite was identified as escherichelin (feature No. *10, [M+H]^+^: C_13_H_11_N_2_O_3_S_2_^+^, calculated: *m*/*z* 307.0206, measured: *m*/*z* 307.0201, Δm −1.5 ppm), a known derivative of Ybt, by means of the fragmentation pattern and comparison with the literature (see Section 2.3). Another known derivative of Ybt named ulbactin B (feature No. *11, [M+H]^+^: C_17_H_21_N_2_O_3_S_2_^+^, calculated: *m*/*z* 365.0988, measured: *m*/*z* 365.0986, Δm −0.6 ppm), which has not been described for *E. coli* before, was identified as well. Because this derivative has never been described in *E. coli* before, ulbactin B was isolated from culture supernatant and its NMR data were compared to literature data to confirm the identity (see Section 2.3). The feature with the *m*/*z* 323.0550 (feature No. *12, abbreviation: **8**) for the [M+H]^+^ could not yet be identified by MS^2^ and MS^3^ experiments. Because of the comparable molecular formula in comparison to Ybt and its derivatives as well as the common occurrence with Ybt, it is assumed that this metabolite could be related to Ybt. Feature No. *16 is assumed to be aerobactin ([M+H]^+^: C_22_H_37_N_4_O_13_^+^, calculated: *m*/*z* 565.2352, measured: *m*/*z* 565.2345, Δm −1.3 ppm) based on the fragmentation pattern (Appendix A), the predicted molecular formula, and the knowledge that the *E. coli* strains EcN, 83972, and CFT073 can produce this siderophore [12,13,14,15]. Appendix A shows the extracted ion chromatograms (EIC, *m*/*z* 565.2345, aerobactin) of the culture supernatant from *E. coli* strains EcN, 83972, and CFT073 and the blank sample. The peak area of aerobactin in the metabolome of *E. coli* CFT073 was approximately 10-fold higher compared to the metabolomes of the other two strains. Reference MS/MS spectra for aerobactin can be found in the literature ([48], in the Appendix A). In contrast, the peak area of aerobactin in the metabolome of EcN was about twice as high compared to the metabolome of *E. coli* 83972. The distribution of feature *16 in the volcano plots reflects these differences in the production of aerobactin, which seems to partially explain the difference in the metabolomes of the three strains regarding principal component 2. Further features of statistical analysis include, for example, the iron complex and the sodium adduct of aerobactin, thus leading to a large number of features, which have their origin in this siderophore, but are not listed here. Due to the fact that this siderophore is known for all three strains, aerobactin will not be discussed further.

The growth of the bacterial strains was monitored by measurement of the optical density (OD) value at 600 nm after cultivation of the different cultures. The average OD value for EcN was 0.491 ± 0.035, for *E. coli* 83972 0.215 ± 0.021, and for *E. coli* CFT073 0.515 ± 0.047. The determined OD values are shown in Appendix A. It is noticeable that the OD values of *E. coli* 83972 are only about half as high as those of *E. coli* strains EcN and CFT073. After cultivation as a static culture, a strong formation of a biofilm was observed in the cultures of *E. coli* 83972 at the bottom of each flask. This was not the case for the other two strains.

### 2.2. Comparison of the Occurrence of Ybt and Several Derivatives in the Metabolome of Different E. coli Strains

The described novel derivatives of Ybt **1**-A (feature No. *6) and **2**-A (feature No. *8) as well as Ybt (features No. *1 and *3) itself and the known derivatives escherichelin (feature No. *10) and ulbactin B (feature No. *11) were detected mainly in the metabolome of EcN and were used to distinguish the metabolomes of the three strains in this study. However, as described above, in the metabolome of *E. coli* 83972, some of the selected features corresponding to the Ybt derivatives were detected as well but with significantly lower intensities compared to EcN. This is in good agreement with the literature, since it is known that *E. coli* 83972 is also capable of biosynthesizing Ybt. However, the amount of the produced derivatives formed during bacterial growth of EcN and *E. coli* 83972 and released into the medium seems to differ significantly under the described cultivation conditions, leading to the differentiation of the metabolomes of the strains.

Due to the higher sensitivity, all bacterial samples were analyzed by triple quadrupole LC-MS/MS to compare Ybt and its derivatives present in the culture supernatant. Since no reference standards were available, only the peak areas of the analytes were compared. All three biological replicates of each *E. coli* strain were analyzed in triplicate and the respective peak areas of the metabolites were averaged. As expected, Ybt and its derivatives were found only in the metabolomes of EcN and *E. coli* 83972. For the comparison of the peak areas, the average peak area in the metabolome of EcN was used as a reference (100%) and the relative peak areas of the respective compounds in the metabolome of *E. coli* 83972 were referenced to that value. In comparison, the amounts of the two Ybt isomers for the *E. coli* strains EcN and 83972 showed significant differences based on the peak areas. Figure 4 shows the LC-MS/MS chromatograms of the culture supernatant from the *E. coli* strains EcN and 83972, displaying the two isomers of Ybt.

The comparison of the relative peak areas corresponding to the metabolites of interest is shown in Figure 5. The *E. coli* strain 83972 showed a relative peak area of 2.7% for the first Ybt isomer and 2.8% for the second isomer compared to EcN. Fe(III)-Ybt exhibited a relative peak area of 2.9%, which is comparable with the relative peak areas of the two isomers of Ybt. Cu(II)-Ybt, on the other hand, showed a higher relative peak area of 19.7%. The chromatograms of the culture supernatant from *E. coli* strains EcN and 83972, displaying the two metal complexes of Ybt, are shown in the Appendix A.

The two known derivatives escherichelin and ulbactin B also showed much lower concentrations in the metabolome of *E. coli* 83972. Here, the relative peak area of escherichelin constituted approximately 11.8% of the peak area compared to EcN and thus showed a slightly higher production of this derivative in comparison with Ybt. The production of ulbactin B seemed to be lower in the culture supernatant of *E. coli* strain 83972 as well, where the peak area was about 1.9% of the relative peak area of ulbactin B in the metabolome of EcN. The respective LC-MS/MS chromatograms of the culture supernatant from *E. coli* strains EcN and 83972, displaying the two derivatives of Ybt, are shown in the Appendix A.

The comparison of the relative peak areas of the novel derivatives **1**-A and **2**-A also implied a similarly low production of Ybt and ulbactin B compared to EcN (Appendix A). In the metabolome of *E. coli* strain 83972, **1**-A showed only about 1.2% and **2**-A about 2.7% of the peak area compared to EcN. These peak area ratios were more comparable to the ratio of Ybt production.

### 2.3. Identification of Escherichelin and Ulbactin B

In this study, the metabolites escherichelin and ulbactin B, known derivatives of Ybt, were detected as discriminating features comparing the metabolome of *E. coli* strains EcN, 83972, and CFT073. The total ion chromatogram (TIC) and the EIC of the pre-concentrated culture supernatant of EcN, displaying escherichelin and ulbactin B next to Ybt, which was obtained during the isolation of ulbactin B, are shown in Figure 6.

Escherichelin was identified by comparing the obtained MS^2^ spectrum (Figure 7) with literature data [29]. The same NL of 87 (fragment a), 46 (fragment b), 78 (fragment d), and 104 Da (fragment e) were observed as already described by Ohlemacher et al. [29]. Additionally, under these conditions, an NL of 44 Da (fragment c) was detected and is caused by a loss of CO_2_, which typically indicates the occurrence of a carboxylic acid group.

Ulbactin B was isolated for the first time from marine bacteria and its structure has already been elucidated; therefore, NMR data were available [31]. However, since no MS^2^ data are available for comparison and ulbactin B has not yet been described for *E. coli*, it was isolated from the supernatant of cultures of EcN for conclusive identification. A total amount of 0.2 mg of isolated ulbactin B was sufficient to obtain ^1^H and ^13^C NMR spectra as well as 2-D NMR spectra, including gradient-selected correlation spectroscopy (gCOSY), gradient-selected heteronuclear single-quantum correlation (gHSQC), and gradient-selected heteronuclear multiple-bond correlation (gHMBC) (see Appendix A). With this set of spectra and comparison to the literature, the identity of ulbactin B was confirmed. Table 2 shows all chemical shifts, which match those reported in the literature. Small differences in the chemical shifts are attributed to the use of different solvents for the respective NMR measurement. In this study, MeOD was used, whereas in the literature, CDCl_3_ was used. These differences are, however, rather small for non-protic protons, allowing for a good comparison of the spectra.

The ^1^H NMR spectrum showed the expected ^1^H signals for ulbactin B. Due to the low amount of the isolated ulbactin B, not all expected ^13^C signals could be observed in the ^13^C NMR spectrum. Especially, the quaternary C atoms were not detectable in the ^13^C NMR spectrum but could be detected in the gHMBC NMR spectrum. The remaining missing ^13^C signals were identifiable by means of ^1^*J*
^13^C ^1^H spin couplings in the gHSQC NMR spectrum. Thus, based on the 2-D COSY spectrum in combination with the gHSQC spectrum, three isolated spin systems were assigned. The first spin system included C1 to C7, the second one C8 to C10, and the third one C11 to C17.

Ulbactin B has not been described to such an extent in the literature, and there are no published fragmentation spectra yet. Figure 8 shows the recorded high-resolution MS^2^ spectrum of ulbactin B and the proposed fragmentation pattern. During the fragmentation of ulbactin B, the NL of 153 (fragment a), 119 (fragment b), 165 (fragment c), 175 (fragment d), and 183 Da (fragment e) were observed. The fragments were comparable to each other and it was assumed that they originate from the center of the molecule between the C7 and C10 atom, especially in or close to the thiazoline ring. The resulting fragment with the *m*/*z* 190.0321 for the [M-175+H]^+^ ion could also be observed during the fragmentation of Ybt (Table 6) and corresponds to the cleavage of the bond of C10 to the neighboring nitrogen and sulfur atom.

### 2.4. Novel Derivatives of Ybt.

During the isolation of ulbactin B, further novel derivatives of Ybt occurring in different isomeric forms, next to the unknown derivatives **1**-A and **2**-A, were detected in the pre-concentrated supernatant of cultures of EcN (Figure 9). **1**-A and **2**-A represent the isomeric forms, which were first detected as some of the main metabolites in the present metabolomics study (feature No. *6 and *8) (see Section 2.2). Due to a low concentration, however, it is assumed that these other unknown derivatives (**1**-B, **2**-B and **3**–**7**) could not be detected during the metabolomics study. As all of the novel Ybt derivatives (**1**–**7**) were only occurring in low concentrations, their isolation and subsequent structure elucidation was not feasible within the scope of the study. Thus, further characterization of these metabolites was only possible based on extensive MS experiments (see below). Additionally, known metal complexes of Ybt with nickel (Ni(II)-Ybt, [M+H]^+^: C_21_H_26_NiN_3_O_4_S_3_ calculated: *m*/*z* 538.0433, measured: *m*/*z* 538.0430, Δm −0.6 ppm) and aluminum (Al(III)-Ybt, [M+H]^+^: C_21_H_25_AlN_3_O_4_S_3_ calculated: *m*/*z* 506.0817, measured: *m*/*z* 506.0805, Δm −2.4 ppm) were determined during the isolation of ulbactin B.

The assumption that the abovementioned unknown metabolites most likely represent structural derivatives of Ybt was based on the considerable similarities observed regarding the mass spectrometric fragmentation behavior of these compounds and Ybt. For nearly all novel derivatives of Ybt, two peaks with the same accurate mass but different fragmentation spectra were detected by LC-(MS/)HRMS, indicating the occurrence of two different isomeric forms each. The TIC and EIC of Ybt and its novel derivatives are shown in Figure 9. In the following, the two isomeric forms of the novel derivatives will be referred to as isomer A and isomer B. The *m*/*z* for the [M+H]^+^ ion of these metabolites and the introduced abbreviations are summarized in Appendix A.

The classification of these isomers is based on the respective MS fragmentation spectra and is further described in the following. An exception is the derivative with the *m*/*z* 508.1391 for the [M+H]^+^ ion: In contrast to the other compounds, three peaks were observed for this metabolite, showing a similar fragmentation behavior. These three isomeric forms are thus labeled isomers A_1–3_. The obtained fragmentation spectra do not allow the prediction of an exact structure, however, as there are several possibilities and positions for the modifications. Table 3 summarizes the predicted molecular formulas for these novel derivatives based on the fragmentation experiments and displays the comparison of their molecular formulas to the one of Ybt.

Since the further structure elucidation of the derivatives is based on the comparison of their fragmentation spectra to the one corresponding to Ybt, Figure 10 shows the main NL and the possible fragments of Ybt, which were obtained upon MS^2^ and MS^3^ experiments and are in compliance with literature data [18]. In order to provide a clear description of the mass spectrometric fragmentation of Ybt, the molecule was divided into three parts and marked with different colors and the letters A, B, and C. The typical NL of 187 Da of part A (fragment a) corresponds to the cleavage of the bond between C13 and C14 during MS^2^. MS^3^ experiments revealed an NL of 105 Da (fragment c), which is caused by cleavage of the thiazolidine ring. For a better comparison of the different structures, the remaining moieties after this cleavage are described as part B and part C. Fragment b results from a cleavage of the thiazolidine ring, while fragment e seems to be specific for the thiazoline ring. The NLs of 88 (fragment b) and 34 Da (fragment e) also occur simultaneously, corresponding to fragment d with an NL of 122 Da. The loss of the sulfur atom in thiazolines (between C7 and C8 in Ybt) was already observed for escherichelin. Regarding the NL of 122 Da, it is assumed that this NL includes the NL of 88 and 34 Da.

#### 2.4.1. Characterization of the A Isomers of the Novel Derivatives of Ybt Based on MS Data

It is not possible to make a precise prediction about the exact structure of the novel derivatives based on the fragmentation spectra. However, in order to propose structural modifications in relation to Ybt, the fragmentation spectra of the derivatives and those of Ybt were compared. For this purpose, only selected fragments of the obtained spectra based on MS^2^, MS^3^, and MS^4^ experiments are described in detail (complete fragmentation spectra can be found in the Appendix A).

The isomeric form A of the derivatives always showed the typical NL for Ybt (187 Da, C_8_H_13_NO_2_S) as their main fragment, which is consistent with part A (Figure 10). An exception is derivative **5**-A, which exhibits an NL of 173 Da (C_7_H_11_NO_2_S) instead of 187 Da, most likely due to the missing methyl group usually located in part A. This NL leads to the main fragment with the *m*/*z* 414.0610 for the [M-173+H]^+^ ion, which also represents the main fragment of **2**-A (*m*/*z* 414.0610 for the [M-187+H]^+^ ion). Table 4 lists the *m*/*z* of the [M+H]^+^ ions, the typical NLs, and the resulting main fragments (part B and C) of the derivatives derived from the fragmentation experiments.

Because of this common loss of part A for most derivatives, it was assumed that the structural modifications are located in the remaining fragment (part B and C). These main fragments were further investigated using MS^3^ experiments. The resulting fragments of the same main NL were compared and are reported in Table 5. Besides these common NLs, the derivatives also showed some identical fragments, which originate from different NLs compared to Ybt (Table 6). Some characteristic fragments as well as NLs are only observable for these compounds employing MS^4^ experiments; a comparison of those is shown in Table 7 and Table 8 as well.

When comparing the molecular formulas of **1**-A and Ybt (Table 3), a difference equal to an additional O atom, probably in the form of an OH group, is assumed as a modification. **1**-A shows an NL of 18 Da (Table 5), which indicates the loss of water, resulting in the occurrence of the fragment with the *m*/*z* 293.0413 for the [M-187-H_2_O+H]^+^ ion. In this case, it is likely that this fragment is the same fragment, which is observed for Ybt after the NL of 187 Da (Table 4), with the *m*/*z* 295.0570 for the [M-187+H]^+^ ion, only with an additional double bond. A loss of water from the main fragment can also be observed for Ybt. Therefore, it cannot be clearly stated at this point whether the water loss detected for **1**-A is caused by an additional OH group. However, a comparison of the water loss between Ybt and **1**-A in the MS^2^ experiments provides a further hint. Here, Ybt shows a water loss with a considerably low abundance, which is hardly visible in the fragmentation spectrum. On the other hand, the cleavage of water from the parent ion of **1**-A is much more abundant. In the case of the water loss starting from Ybt, the OH group at C13 could be involved (Figure 10). However, the cleavage of water at this position does not seem favored due to steric interference. Firstly, there is no H atom at C14 for the formation of water. Furthermore, the formation of hydrogen bonds between the OH group on C13 and the neighboring nitrogen atom between C10 and C12 is conceivable. This could also result in a stabilized system, which would make the cleavage of water at this position less likely. Since, in contrast to Ybt, the water loss of **1**-A resulted in a higher intensity of the formed fragment, this cleavage appears to be more prevalent. In conclusion, it is assumed that the structures of Ybt and **1**-A are similar except for the additional OH group of **1**-A.

The derivatives **2**-A as well as **5**-A are suspected to contain a cysteine group as side chain, since an NL of 121 Da (C_3_H_7_NO_2_S) (Table 6) was observed for both. Due to a further NL of 87 Da (C_3_H_5_NO_2_, the corresponding fragment with the *m*/*z* 327.0290 for the [M-187-87+H]^+^ or [M-173-87+H]^+^ ion, C_13_H_15_N_2_O_2_S_3_^+^, Appendix A), which corresponds to the molecular formula of cysteine without H_2_S, it is assumed that the bond between the carbon and sulfur atom of the assumed cysteine side chain is cleaved. This would imply that the cysteine group is linked to the Ybt core structure via the sulfur atom. However, this assumption cannot be conclusively proven based on the fragmentation spectra. A different interpretation of the spectra is possible with the NL of 87 Da potentially caused by the loss of an alanine group, and the whole modification of the molecule, thus consisting of an additional alanine as well as an SH group.

The derivative **3**-A could carry a methylated cysteine side chain due to the observed NLs of 133 (C_4_H_7_NO_2_S) and 135 Da (C_4_H_9_NO_2_S) (Table 6). However, a linkage via the sulfur atom or the oxygen atom of the possible carboxy group cannot be proven, since no fragment was observable without the sulfur atom, as is the case for **2**-A. After cleavage, **3**-A also shows the NLs of 133 and 135 Da, resulting in the fragments with *m*/*z* 295.0570 for the [M-187-133+H]^+^ ion as well as 293.0413 for the [M-187-135+H]^+^ ion. The fragmentation caused by the loss of the side chain could either result in a ring closure of the cleaved side chain enabling the NL of 133 Da, or in the formation of an open chain fragment (NL 135 Da) and the formation of a double bond in the resulting ion. Assuming a similar modification of the derivatives **2**-A and **3**-A, a methylation could have taken place either at the amino or carboxy group of the cysteine side chain, leading to the formation of a thiazolidine ring from the leaving group.

When comparing the fragmentation spectra of **2**-A and **4**-A, it can be assumed that these two derivatives are also quite similar and only differ in an additional sulfur atom in the case of **4**-A (Table 3). The important NLs and fragments of these two derivatives, which are assumed to include the modification of an added cysteine group compared to Ybt, are summarized in Table 9.

Like **2**-A, **4**-A shows the NLs of 87 (C_3_H_5_NO_2_) and 121 Da (C_3_H_7_NO_2_S) (Table 9), which have already been described to potentially correspond to an alanine and cysteine group, respectively. Furthermore, both derivatives also show the NL of 153 Da (C_3_H_7_NO_2_S_2_). This NL could be a loss of cysteine, which contains an additional double bond (C_3_H_5_NO_2_S), accompanied by a loss of H_2_S. All fragments formed due to the described NLs of **4**-A contain an additional sulfur atom based on the predicted molecular formulas compared to **2**-A. The loss of 119 Da (C_3_H_5_NO_2_S), which could correspond to the cysteine group with an additional double bond, was only observed for **4**-A. This led to the assumption that the cysteine side chain is possibly linked to the core structure of Ybt via a disulfide bridge in the case of **4**-A, which enables the homolytic cleavage of this disulfide bond.

Regarding the derivatives **1**-A, **2**-A, **3**-A, **4**-A, and **5**-A, the NL of 88 Da (C_3_H_4_OS) (Table 5 and Table 7 and Table 8) is observable in all MS^3^ and MS^4^ experiments. This loss takes place in part B of the molecules, probably upon cleavage of the bonds between C12 and the nitrogen atom as well as C10 and the sulfur atom (Figure 10). As this NL is also detectable for Ybt, it can be assumed that none of the modifications of the novel derivatives are located at this part of the molecule.

The NL of 105 Da (C_3_H_7_NOS) (Table 6), shown for Ybt and leading to the formation of part C (Figure 10), was not observed for **2**-A, **3**-A, **4**-A, and **5**-A. This loss may include the NL of 88 Da, where no modification is assumed, and results from the cleavage of the bonds between C10 and the neighboring nitrogen and sulfur atoms. As the derivatives do not show this cleavage, it is assumed that the structural modification is close to the C10 atom or that the additional moiety is directly linked to it. This influences the fragmentation behavior of the derivatives, preventing the respective neutral loss of 105 Da. Disregarding the unlikely possibility of radical formation, the loss of the modifications (cysteine side chain for **2**-B and **5**-B, methylated cysteine side chain for **3**-B, cysteine group with an additional sulfur atom for **4**-B) is only possible accompanying the formation of a double bond in the remaining fragment; this strongly hints towards the localization of all modifications at either the C8, C9 or C10 atom.

**1**-A, on the other hand, shows an NL of 103 Da (C_3_H_5_NOS) (Appendix A), resulting in the presence of the fragment with the *m*/*z* 208.0427 for the [M-187-103+H]^+^ ion (C_10_H_10_NO_2_S^+^). Here, the same part of the molecule as for the NL of 105 Da (part B), which was described for Ybt before (Figure 10), is assumed to be cleaved but with the formation of an additional double bond. This derivative hence shows the loss of part B compared to the other derivatives, but in a slightly different manner as observed for Ybt. It is assumed that **1**-A exhibits the additional modification at either the C8, C9, or C10 atom, which apparently influences the fragmentation behavior. This difference in the fragmentation behavior might be due to **1**-A having an additional free OH group compared to Ybt, while the derivatives **2**-A, **3**-A, **4**-A, and **5**-A differ from Ybt by an additional side chain, which likely consists of cysteine. Another possibility is that the position of the modification for **1**-A could be attached to a different C atom, for example, C10, because of the described NL of 103 Da compared to the already mentioned derivatives.

The fragment with the *m*/*z* 173.0709 for the [M-187-122+H]^+^ ion is formed after the NL of 122 Da upon fragmentation of Ybt (Table 5 and Table 6), and is part of part C. This NL is likely a combination of the previously explained loss of 88 (C_3_H_4_OS) and 34 Da (H_2_S) starting from the remaining thiazoline ring. The NL of 34 Da (H_2_S) is observable for escherichelin as well (Figure 7). The resulting fragment with the *m*/*z* 173.0709 was also observed for **2**-A, **3**-A, **4**-A, and **5**-A (Table 6 and Table 7), which is likely be formed due to the NL of 122 Da and the respective modification “X” as the [M-187-122-X+H]^+^ ion. **1**-A, on the other hand, shows no fragment with the *m*/*z* 173.0709. However, the fragment with the *m*/*z* 174.0544 (C_10_H_8_NO_2_^+^) can be observed for the [M-187-137+H]^+^ ion after the NL of 137 Da (C_3_H_7_NOS_2_) (Appendix A). This NL of 137 Da is potentially resulting from the losses of 103 (C_3_H_5_NOS) and 34 Da (H_2_S). Consequently, the additional OH group of **1**-A would still be part of the fragment, implying that this structural modification is located in part C. This supports the assumption that the modification is likely located between the C8, C9, and C10 atom.

Another indication that the OH group of **1**-A may be attached to the C10 atom is the observed NL of 133 Da (C_4_H_7_NO_2_S), which results in the formation of the fragment with the *m*/*z* 178.0319 for the [M-187-133+H]^+^ ion (C_9_H_8_NOS^+^) (Appendix A). This NL is also observable for **3**-A, but the NL for this derivative indicates a modification that is suspected to correspond to a methylated cysteine unit. In the case of Ybt, on the other hand, a fragment with the *m*/*z* 178.0319 can be observed as well and described as the [M-187-117+H]^+^ ion with an NL of 117 Da (C_4_H_7_NOS) (Appendix A). This NL appears to correspond to part B and the adjacent C10 atom. In comparison, the NL of 133 Da for **1**-A includes an additional oxygen atom. In combination with the already discussed NLs of 88 and 103 Da located in part B, the OH group is likely attached to the C10 atom in the case of **1**-A.

The derivative **6**-A also showed a water loss during the MS^n^ experiments, which exhibited a higher intensity than the water loss observed for Ybt (Table 5 and Appendix A). This may be caused by an additional OH group of this derivative as well. During the MS^3^ experiments, **6**-A showed a CO loss starting from the main fragment with the *m*/*z* 343.0241 for the [M-187+H]^+^ ion, including part B and C, which in turn is assumed to include the C13 atom of part B (Appendix A). Additionally, this derivative does not show the described typical loss of 88 Da originating from part B. Therefore, it is likely that the structure at C11 or C12 is slightly modified. Similarly, a loss of 135 (C_3_H_5_NOS_2_) and 165 Da (C_4_H_7_NO_2_S_2_), which led to the fragments with the *m*/*z* 208.0425 (C_10_H_10_NO_2_S^+^) for the [M-187-135+H]^+^ ion and 178.0314 (C_9_H_8_NOS^+^) for the [M-187-165+H]^+^ ion (with lower intensity), were observed in the MS^3^ experiments (Appendix A). These NLs were not observed for Ybt or the other derivatives; however, the resulting fragment with the *m*/*z* 208.0425 was detectable for **1**-A (for the [M-187-103+H]^+^ ion) and the fragment with the *m*/*z* 178.0314 for **1**-A ([M-187-133+H]^+^ ion; lower intensity) and Ybt ([M-187-117+H]^+^ ion) as mentioned before. In comparison to **1**-A, the NLs described for **6**-A (135 Da and 165 Da) differ by one additional sulfur atom compared to the NLs that lead to the same fragments for **1**-A (103 and 133 Da). As the resulting fragments with the *m/z* of 208.0425 (C_10_H_10_NO_2_S^+^) and 178.0314 (C_9_H_8_NOS^+^) likely include part C, the NLs leading to their formation are assumed to include C11 to C13 or C10 to C13, respectively. Thus, the NL of 135 Da hints toward the sulfur atom being attached to either the C11 or C12 atom in the form of an SH group, or to be included in the ring in the case of **6**-A. Based on the NL of 165 Da, an OH group could be linked to the C10 atom, as was already suspected for **1**-A. However, this conclusion is only a presumption and can neither be proven nor further elucidated solely based on the fragmentation spectra.

The derivative **7**, which occurs in three isomeric forms, each showing the same fragmentation behavior, seems to have a different structure compared to the other derivatives. Due to the NL of 187 Da considered to be the typical NL of Ybt (Table 4), and several fragments that were also observed for Ybt and partly other derivatives, this derivative is suspected to show a structural similarity to Ybt. However, a different type of modification is assumed though, as the retention time (Figure 9) and the fragmentation behavior considerably differ from the other derivatives (Appendix A). Comparing the molecular formulas of this derivative and Ybt, the difference of C_2_H_2_ becomes apparent (Table 3). Despite many fragments, however, no conclusive structural proposal can be derived from the obtained data. It can be assumed that the modification is located in part B. Part A is not modified as evidenced by the NL of 187 Da. Similarly, part C is unlikely to be modified, since common fragments for part C compared to Ybt were observed, for example, the fragment with the *m*/*z* 190.0319 (C_10_H_8_NOS^+^) (Table 6).

#### 2.4.2. Characterization of the B Isomers of the Novel Derivatives of Ybt Based on MS Data

The B isomers differ from the A isomers in that they show no main NL of 187 Da, indicating a difference in part A of the molecules; in general, most of the obtained fragments do not show any similarity to those observed upon fragmentation of Ybt. For the derivatives **2**-B, **3**-B, **4**-B, and **5**-B, main fragments with the *m*/*z* 365.0986 for the [M-X+H]^+^ ions are detectable, which correspond to *m*/*z* observed for ulbactin B and also show the same fragmentation during MS^3^ experiments. This kind of fragmentation was also apparent for Ybt (Appendix A), where it is assumed that an oxygen of the carboxyl group causes a ring formation at the core structure. Due to the low intensity of the resulting fragment, however, this fragmentation reaction does not seem to be preferred for Ybt. In contrast, the derivatives **1**-B and **6**-B do not show a fragment with the *m*/*z* 365.0986 for the [M-X+H]^+^ ion. The respective NLs leading to the main fragment of the derivatives are listed in Table 10.

The following Table 11 and Table 12 show the chosen NLs and fragments of the respective B isomers of the derivatives with the exception of **6**-B. The corresponding fragmentation spectra are displayed in the Appendix A. The derivative **6**-B shows a completely different fragmentation behavior compared to the other derivatives. In the case of **6**-A, the presence of an OH and an SH group was already proposed. Whether this isomer also has an OH and SH group attached to the core structure of Ybt can only be presumed. Several common fragments are observable in comparison to the fragmentation of Ybt, which still leaves the assumption that it could be a derivative of Ybt. For example, the fragmentation spectrum of **6**-B shows fragments with the *m*/*z* 173.0708 for the [M-357+H]^+^ ion and with the *m*/*z* 261.0696 for the [M-269+H]^+^ ion (Appendix A); these were also observed upon fragmentation of Ybt. However, since the fragmentation spectrum is considerably different from those of the other derivatives, no conclusive statement can be made here about whether the core structure of Ybt still remains unchanged, as is assumed for the other derivatives. For this reason, no further assumptions about the structure of this derivative can be made and therefore this derivative was excluded from the following tables.

In comparison to its isomer **1**-A, **1**-B also shows the cleavage of the bond between the C13 and C14 atom of the Ybt core structure, which corresponds to the loss of part A (NL of 187 Da). However, this cleavage does not seem to be preferred for the B isomer, since the fragments formed exhibited low intensities (Appendix A). A different charge distribution can also be observed, with the part of the molecule yielding an NL of 187 Da for the A isomer retaining the positive charge and thus being detected as a fragment with the *m*/*z* 188.0740 for the [M-310+H]^+^ ion (Table 12). Due to this different fragmentation behavior, a different structure is assumed for the B isomer, which influences the distribution of the charge.

The NL of 120 Da (C_7_H_4_O_2_) results in the formation of the main fragment of **1**-B with the *m*/*z* 378.0975 for the [M-120+H]^+^ ion (Table 10). Due to the high number of double bond equivalents of the neutral fragment corresponding to this NL, it is assumed that this is the aromatic region of the Ybt structure of part C (C1 to C7). The predicted molecular formula indicates an additional OH group in comparison to Ybt. It is conceivable that the additional OH group is located at the C7 atom or at the aromatic ring. The linkage of the OH group to position C7 would likely be supported by the occurrence of the NL of 120 Da, which was not observed for **1**-A. Due to the change of the structure, the cleavage of C_7_H_4_O_2_ is preferred. However, the double bond between C7 and the neighboring nitrogen atom would also have to be shifted under this assumption. On the other hand, an additional OH group on the aromatic ring, bound in the *ortho* position to the already existing OH group, is conceivable. This could also lead to a loss of water, which was also observed for **1**-B.

The fragment with the *m*/*z* 480.1079 for the [M-H_2_O+H]^+^ (Table 11) ion is formed starting from **1**-B due to a loss of water, which leads to the core structure of Ybt and an increase in the double bond equivalents, likely caused by the formation of a double bond here. It is assumable that this water loss represents the corresponding modification of the additional OH group of this derivative compared to Ybt, as was the case for the A isomer. A water loss originating from the fragment with the *m*/*z* 464.1129 for the [M-H_2_O+H]^+^ ion was also observed for Ybt as described above, although with a very low intensity (Appendix A). Since the water loss in **1**-B is more abundant compared to Ybt, as is the case for **1**-A, it is more likely that this loss could be caused by the additional OH group.

In the case of the derivatives **2**-B, **3**-B, and **4**-B, the fragment described with the *m*/*z* 480.1079, each formed due to different NLs, can also be observed (Table 11). These NLs correspond to the respective modification of the derivatives compared to Ybt, which was observed for the A isomer. Therefore, it is assumed that the NL of the respective fragments observed is caused by the respective modification and that the modification is the same as for the respective A isomer. Thus, **2**-B shows a loss of cysteine corresponding to an NL of 121 Da, as observed for **2**-A. **3**-B shows an NL of 135 Da, which was already suspected to be caused by a loss of methylated cysteine in the case of the A isomer. The NL of 153 Da detected for **4**-B corresponds to a cysteine group with an additional sulfur atom. Here, it was assumed that the cysteine unit of the A isomer was already linked to the Ybt core structure via a disulfide group. Compound **5**, which differs from **2** only in the absence of a methyl group, also shows the loss of cysteine (121 Da) for the B isomer, leading to the fragment with the *m*/*z* 466.0921 for the [M-121+H]^+^ ion. Thus, **5**-B shows the same type of modification as **5**-A. The double bond equivalents also indicate the formation of a double bond or a ring in the core structure after loss of the modification for all derivatives, as was the case for the A isomer.

For **1**-B and **2**-B, a fragment with the *m*/*z* 291.0832 (C_11_H_19_N_2_O_3_S_2_^+^) (Table 12) is observed in each case. Herein, it is assumed that it represents part A and B after the loss of part C, which implies a cleavage of the bond between C10 and the neighboring nitrogen and sulfur atoms (Figure 10). This type of cleavage was also observed and described for Ybt, leading to the fragment with the *m*/*z* 190.0320 (C_10_H_8_NOS^+^) (Table 5 and Table 6), which consists of part C. In the case of **1**-B, the fragment with the *m*/*z* 291.0832 is caused by the NL of 207 Da (C_10_H_9_NO_2_S). Compared to the fragment described for Ybt (C_10_H_8_NOS^+^) indicating part C, it could represent the same part of the molecule but as an uncharged NL and with an additional oxygen atom. This implies the existence of an additional OH group in part C of the molecule as described above. Nevertheless, it is also possible that part C and a water unit are cleaved off as two separate molecules, so that the linkage of an OH group to the C7 atom, for example, would not be mandatory. In the case of **1**-B, no loss of part C was observed without the loss of water. This further suggests that the OH group is likely attached to C7. With regard to **2**-B, the NL of 310 Da (C_13_H_14_N_2_O_3_S_2_) results in the formation of the fragment with the *m*/*z* 291.0832. For this NL, it is assumed that the same bonds described for Ybt and **1**-B are cleaved and thus the same part of the molecule is affected (part C with C_10_H_7_NOS, 189 Da). The cleavage at C10, which was not observed for **2**-A, could again confirm that the modification is positioned elsewhere in **2**-B. However, it is not clear based on this NL whether the additional assumed cysteine side chain (C_3_H_7_NO_2_S, 121 Da) is still bound to part C with C_10_H_7_NOS (189 Da) or whether two separate NLs occur, such as cysteine and C_10_H_7_NOS (189 Da).

For the derivatives **2**-B, **3**-B, **4**-B, and **5**-B, a structural modification of the part of the molecule, which prevents the NL of 187 Da (part A) typical for Ybt, is assumed, as none of these derivatives show this characteristic NL. If the difference between the molecular formulas of the respective modifications described above (cysteine side chain for **2**-B and **5**-B, methylated cysteine side chain for **3**-B, cysteine group with an additional sulfur atom for **4**-B) and the NL from Table 10, which leads to a main fragment similar to ulbactin B upon further fragmentation, are considered for the derivatives **2**-B, **3**-B, and **4**-B, the difference of C_4_H_5_NOS always becomes apparent. In the case of **2**-B, for example, the NL, which results in the formation of the main fragment, corresponds to 236 Da (C_7_H_12_N_2_O_3_S_2_), and the presumed modification is a cysteine side chain (C_3_H_7_NO_2_S), the difference in the predicted molecular formulas amounts to C_4_H_5_NOS. Thus, it is assumed that the respective modification could be located adjacent to a group with this difference of C_4_H_5_NOS. Based on the structure of Ybt, the NL of C_4_H_5_NOS could originate from the carbon atoms C17 to C19 and the bound nitrogen and sulfur atom of part A. In the case of **5**-B, a difference in the molecular formulas of C_3_H_3_NOS between the NL, which forms the main fragment (C_6_H_10_N_2_O_3_S_2_, 222 Da), and the presumed modification of a cysteine side chain (C_3_H_7_NO_2_S, 121 Da), is observed. This could indicate the missing methyl group at C17 compared to the other derivatives, especially **2**-B, because they seem to have the same modification. Due to rearrangements upon fragmentation, an oxygen atom of the carboxyl group could lead to the formation of a structure alike to ulbactin B as the main fragment. However, it cannot be conclusively derived here whether the described NL shown in Table 10 for **2**-B, **3**-B, **4**-B, and **5**-B, which includes the respective modification in combination with C_4_H_5_NOS for **2**-B, **3**-B, and **4**-B or C_3_H_3_NOS for **5**-B, consist of a single part of the molecule or of two separate groups, which are localized on different parts of the core structure. As this NL is very abundant and no NLs, which only contain C_4_H_5_NOS or C_3_H_3_NOS, are observable, it can be assumed that it is rather a bound moiety. Thus, it can be hypothesized that the modification is located at C16.

If the respective modifications of **2**-B, **3**-B, and **4**-B were cleaved off, the number of double bond equivalents could not be explained by the formation of a double bond upon fragmentation due to the methyl group at C17. At this point, it is possible that a six-membered ring with the methyl group at C19 and a double bond is formed during the fragmentation as a result of the modification. Since the cleavage of the respective modification can only be observed with very low intensities of the resulting fragments compared to the formed main fragment similar to ulbactin B, the cleavage reaction does not seem to be preferred. It rather seems that the cleavage of the bond between C15 and the neighboring nitrogen and sulfur atoms occurs preferably if a linkage of the modification to C16 is evident, primarily leading to a fragment similar to ulbactin B.

The derivative **3**-B shows a fragment with the *m*/*z* 295.0563 as [M-320+H]^+^ ion upon fragmentation (Table 12). The predicted sum formula corresponds to that of the main fragment of Ybt (Table 4), which is occurring after the NL of 187 Da and corresponds to part B + C (Figure 10). Accordingly, the NL of 320 Da contains the modification assumed to be methylated cysteine in this derivative and the typical NL of 187 Da. It cannot be concluded whether the loss is caused by the cleavage of a single moiety or if it consists of two fragments. If a single moiety is assumed to be cleaved during this NL, a modification of the part of the molecule, which is typically cleaved resulting in the loss of 187 Da, is plausible.

## 3. Discussion

In this metabolomics study of *E. coli* culture supernatants, it was shown by the application of LC-HRMS that the metabolome of EcN differed significantly from the metabolome of *E. coli* strains 83972 and CFT073 after cultivation in MEM for 6 h at 37 °C without shaking. The metabolomes of *E. coli* CFT073 and 83972 differed only insignificantly. The comparison showed that under the selected cultivation conditions, the metabolomes of the three closely related *E. coli* strains did not show significant qualitative but significant quantitative differences with respect to the detected components. Interestingly, the three isolates did not differ in the basic composition of their metabolomes, but only in the extent to which the metallophore Ytb or the siderophore aerobactin could be detected in the culture supernatant.

The production of metallophores improves the bacterial adaptation to changing microenvironments and thus the bacterial fitness [21,49]. Metallophores function as metal ion-capturing systems, which provide these metal ions required as critical co-factors of enzymes [10,50]. Additionally, they may also be involved in the regulation of bacterial metabolism and signaling [12,29,51,52]. Importantly, metallophores have diverse virulence-associated effects, and they promote bacterial pathogenicity [53,54,55].

Based on the fragmentation patterns, the characteristic isotope patterns, and the predicted molecular formulas, the metallophore Ybt, its known metal complexes with iron and copper, and its known derivatives escherichelin and ulbactin B were identified [29,31]. Interestingly, novel Ybt derivatives with the *m*/*z* 498.1178 (**1**-A) and 601.1270 (**2**-A) for the [M+H]^+^ ion were detected and characterized by LC-MS/HRMS as well and seem to play a role in the differentiation of the metabolomes of the strains. In the metabolome of *E. coli* 83972, some of the Ybt derivatives were detected but with significantly lower intensities than in the case of EcN. It is known that both *E. coli* strains EcN and 83972 are able to synthesize Ybt [13,14], whereas UPEC strain CFT073 has lost the ability to functionally express the Ytb determinant [56,57]. Due to the observation that Ybt and its different derivatives were responsible for clear differences in the metabolome under the applied cultivation conditions, especially in distinguishing the metabolome of EcN from that of the other two *E. coli* strains 83972 and CFT073, Ybt was in the main focus of this study.

During the analysis of the culture supernatants using triple quadrupole LC-MS/MS, all features identified here as Ybt and its derivatives were detected in the metabolome of EcN and, to a much lesser extent, in *E. coli* 83972 as well, due to the high sensitivity of the applied method. Because reference standards were not available for almost all the compounds of interest, only the peak areas could be compared. In the metabolome of *E. coli* strain 83972, Ybt showed only about 2.8%, Fe(III)-Ybt about 2.9%, Cu(II)-Ybt about 19.7%, escherichelin about 11.8%, ulbactin B about 1.9%, and **1**-A about 1.2% and **2**-A about 2.7% of the respective peak areas measured in the metabolome of EcN. Our data demonstrate that *E. coli* 83972 can produce all Ybt-related metabolites but to a significantly lower extent than EcN.

This may explain the results of the statistical analysis. The significantly lower production of the metabolites in the metabolome of *E. coli* 83972 could be the reason for the small difference of the metabolomes of *E. coli* 83972 compared to CFT073. This could also be an explanation for the lower statistical significance of the features in the volcano plot of the *E. coli* strains EcN vs. 83972 in contrast to EcN vs. CFT073. The significant differences in the metabolome of EcN compared to the other two strains probably originate mainly from the considerably higher production of Ybt and its derivatives compared to *E. coli* 83972. Additionally, the higher production of aerobactin by *E. coli* CFT073 compared to EcN and *E. coli* 83972 contributes to the differentiation of the metabolomes as well. Thus, Ybt and its seemingly wide range of derivatives were found to be significant metabolites from the cultivation of these *E. coli* strains in MEM for 6 h. This opens up the question why EcN produces these metabolites in such a higher degree in comparison with *E. coli* 83972.

One reason for these different Ybt concentrations detected could be related to the different growth behavior of *E. coli* 83972 relative to the two other strains (OD values of all samples have already been shown in Section 2.1 and Appendix A). On average, the *E. coli* strain 83972 culture reached a markedly lower OD than the two *E. coli* strains EcN and CFT073. At the same time, *E. coli* 83972 showed a stronger biofilm formation at the bottom of the flask than the other two strains. This biofilm formation can explain the lower bacterial cell density measured for planktonic *E. coli* 83972. Whether the increased biofilm formation of *E. coli* 83972 relative to strain EcN is responsible for the lower Ybt concentration in the culture supernatant remains to be verified. Interestingly, the Ybt receptor FyuA has been shown to be required for efficient biofilm formation, but the mechanism behind (simply via Fe(III)-dependent regulation of gene expression or by a possible contribution as an adhesin) is still unclear [58]. 

The assumption that growth in a static culture in MEM differentially affected gene expression and thus leading to increased biofilm formation of *E. coli* strain 83972 in contrast to EcN and *E. coli* CFT073 was confirmed when the three strains were grown statically in pooled human urine. Upon static growth in pooled human urine, all three strains displayed similar biofilm formation and similar OD values of the cultures (average OD value: EcN −0.174 ± 0.002, *E. coli* 83972 −0.193 ± 0.006, *E. coli* CFT073 −0.157 ± 0.001). In human urine, the Ybt production of *E. coli* strains EcN and 83972 was similar and only escherichelin could be detected. Ulbactin B was only observed in the metabolome of EcN in very low levels after 16 h of cultivation in urine and the novel Ybt derivatives were not detectable. That the growth medium is an important factor for biofilm production has already been shown [44,59].

Various derivatives and structurally similar metabolites of the well-studied siderophore Ybt occur naturally. Numerous derivatives have already been described but for different bacterial genera and species, which further illustrates the diversity of Ybt derivatives produced [30]. The present study indicates that there is a much wider range of derivatives of Ybt than previously known. The derivatives detected in our study were all produced by *E. coli* strains EcN and 83972. This suggests a relation between the synthesis of the derivatives and the biosynthesis of Ybt. Whether other *E. coli* strains producing Ybt or all bacterial species, which are able to synthesize Ybt, are also capable of producing all the derivatives described under these cultivation conditions, especially with regard to the culture medium, is not clear and has to be investigated further.

Regarding the Ybt biosynthetic pathway, escherichelin has recently been described as a physiologically relevant derivative of Ybt, which is produced by many uropathogens [29]. In our study, escherichelin was also identified and detected by LC-MS/HRMS. Another interesting discovery during this study was the detection of ulbactin B, which is a known derivative of Ybt, first isolated from *Vibrio* sp. [31], but which has not yet been described for *E. coli*. Furthermore, we showed for the first time that the *E. coli* strains EcN and 83972 can also produce ulbactin B under these cultivation conditions. As escherichelin, the new *E. coli* metabolite ulbactin B could thus also be a product of the Ybt biosynthetic pathway, since Ybt was also determined in *Vibrio* sp. where ulbactin B was first isolated from. This assumption was described in the literature before. Pre-piscibactin and piscibactin, which are structurally very similar to ulbactin B or Ybt, respectively, were isolated from *Photobacterium damselae* subsp. *piscicida*. Pre-piscibactin was suspected to be an intermediate of the biosynthesis of the siderophore piscibactin. Due to the similar biosynthesis of Ybt and piscibactin as well as their comparable structures, it was already hypothesized that ulbactin B could be a so-called pre-yersiniabactin [60]. Since we detected ulbactin B in both Ybt-producing *E. coli* strains EcN and 83972, even if in very different amounts, our data suggest that ulbactin B is a product of the biosynthesis of Ybt, and that under certain conditions all Ybt-producing bacteria are able to produce this metabolite. However, the conditions under which ulbactin B is produced have to be investigated further. As already mentioned, ulbactin B was detected in very small amounts in the metabolome of EcN after cultivation for 16 h in urine. Further investigations are also needed to determine which role ulbactin B may have for the bacteria and whether it is produced during UTI.

In this study, several previously unknown derivatives of Ybt were detected. The initial suspicion that there are two isomeric forms, which may be diastereomers as is the case for Ybt, could not be confirmed by the fragmentation spectra. The derivatives could only be further investigated by means of various MS^2^, MS^3^, and MS^4^ experiments using LC-HRMS, but a full structure elucidation is not possible and is therefore only proposed.

The Ybt derivatives **1**, **2**, **3**, **4**, **5**, and **6** occur in two isomeric forms. Due to fact that their fragmentation spectra differ not only in relative intensities but also show different fragments, they are likely no diastereomers, notably when compared to the isomers of Ybt, which show fragmentation spectra very similar to each other. Therefore, it is likely that the modifications are located on different parts of the Ybt core structure. The respective isomers of a derivative appear to have the same type of modification based on their fragmentation behavior. Therefore, to propose a possible structure for the derivatives, it is assumed that the type of modification is the same in each case and is only different in the position of the binding to the core structure. An exception is the derivative **7,** which shows three isomeric forms with similar fragmentation spectra.

The isomers of the derivatives were named either isomer A, which always shows the typical NL of 187 Da, or isomer B. Based on the fragmentation spectra and the predicted molecular formula, it is proposed that the derivative **1** has an additional OH group compared to Ybt. For the derivatives **2** and **5**, a cysteine moiety, potentially linked via the sulfur atom to the Ybt core structure, is assumed. A modification, which consists of an alanine group and a separate SH group, is conceivable as well. However, since binding of the alanine moiety via the methyl group is unlikely and a linkage formed by either the carboxyl or amino group would likely show two different fragments depending on whether the hetero atom is part of the NL or not, such a structure is less likely [61]. Furthermore, since Ybt is built up from three cysteine molecules during biosynthesis [21], cysteine is assumed as a likely modification. In addition, the NL of 121 Da (C_3_H_7_NO_2_S) and of 87 Da (C_3_H_5_NO_2_) occur during the mass spectrometric fragmentation, so it is plausible that the cysteine group may be linked via the sulfur atom to the Ybt core structure. The derivative **3** seems to be very similar to **2**. Supposing that the modification of those two derivatives is closely related and based on the fragmentation behavior, which shows a ring closure of the cleaved side chain and an open chain fragment, it is assumed that the modification is a cysteine side chain as well, which might be methylated at the amino or carboxy group, allowing for the formation of a thiazolidine ring. Similarities are also observed in the case of **2** and **4**. They often show a difference of one sulfur atom in the NL or fragments observed during the fragmentation experiments, but there is no conclusive indication where this sulfur atom might be located. Nevertheless, based on the detected NLs, a cysteine side chain, which is linked to the core structure via a disulfide group, is suspected here. For **6**, an additional OH and SH group are proposed. As for **7**, an explicit type of modification cannot be assumed, it can only be derived from the obtained spectral data that this derivative differs from Ybt by C_2_H_2_ and that the modification is probably located in part B but further structural details are unknown so far.

By comparing the fragments of these derivatives, the position of the modifications can be narrowed down. For **1**-A, **2**-A, **3**-A, **4**-A, and **5**-A, it is proposed that the modification is located at C8, C9, or C10. Because the NL of 105 Da, which is caused by the cleavage of C10 from the neighboring sulfur and nitrogen atoms for Ybt, it was not detected for the derivatives, C10 is likely to carry the modification. For **6**-A, an SH group could be attached to C11 or C12 and an OH group to C10, as it is also the case for **1**-A, due to the detected NL of 135 and 165 Da, which may include the part B and C10 of the Ybt structure.

The comparison of the fragmentation spectra of **2**-A with **5**-A and **2**-B with **5**-B allows the presumption that these two derivatives could have the same structure except for the absence of a methyl group for **5**. In general, two positions, in which the methyl group might be absent, are most plausible. It could be possible that one methyl group is missing on C14 because these methyl groups were added during the Ybt biosynthetic pathway by the methyltransferase domain 1 (MT_1_) located on the encoding high-molecular-weight-protein 1 (HMWP1). However, a missing methyl group on C17, which was formed during the biosynthesis by MT_2_, also located on HMWP1, is also possible [21]. Based on the fragmentation spectra of the isomer A, no further assumption can be made, from which methyl group might be missing. However, a comparison of the fragmentation spectra of isomer B allows the hypothesis that the methyl group on C17 is missing based on the comparison of the NLs of 236 Da (C_7_H_12_N_2_O_3_S_2_) for **2**-B and 222 Da (C_6_H_10_N_2_O_3_S_2_) for **5**-B. These NLs are assumed to contain the cysteine side chain as the modification as well as C16 to C19 (**2**-B) or C16 to C18 (**5**-B), including the sulfur and the nitrogen atom. Here, it is assumable that the same methyl group is missing in isomer A and B.

Figure 11 summarizes the presumed positions of the modifications of the derivatives **1**, **2**, **3**, **4**, and **6**-A. For **6**-B and **7**, additional specific positions for the structure based on the fragmentation spectra cannot be given, so these two derivatives are not included. For **5**, the presumed structure is not shown because it is assumed that this derivative differs only in a missing methyl group on position C17 and otherwise has the same modification on the same position as **2**.

In conclusion, a large number of novel Ybt derivatives were detected. The A isomers of the derivatives **1** and **2**, which were already detected during the metabolomics study, seem to be formed preferentially due to their apparently increased occurrence. We suspect that the production of the novel derivatives is directly linked to the production of Ybt. Nevertheless, it appears that modification does not occur in both diastereomers of Ybt because the two isomers of the derivatives do not seem to be diastereomers and the modification is located at different positions in the molecule. However, it is still unclear which function these Ybt derivatives may have for the bacteria and how the different modifications of the Ybt core structure are coordinated and implemented. The corresponding metal complexes, which would further support the idea that the derivatives could also function as metallophores, have not yet been detected. Due to their low concentration and the sensitivity of the applied analytical method, their existence cannot be completely excluded. Recently, another metallophore, designated as HPTzTn-COOH, has been identified to be encoded by the Ybt gene cluster in many UPEC isolates [62], suggesting that enterobacteria can express a multitude of structurally related metabolites with potentially different functions.

## 4. Materials and Methods 

### 4.1. Chemicals and Reagents

All solvents used in this study were of LC-MS grade and purchased either from Carl Roth (Karslruhe, Germany), Thermo Fisher Scientific (Dreieich, Germany) or Sigma-Aldrich (Steinheim, Germany), if not stated otherwise. The used ASTM type 1 water was produced with a Purelab Flex 2 system (Veolia Water Technologies, Celle, Germany). Formic acid (FA) was purchased from Merck (Darmstadt, Germany). Minimum essential medium (MEM) without phenol red was obtained from Thermo Fisher Scientific (Dreieich, Germany). L-alanyl-L-glutamine (200 mM), non-essential amino acids (100x-concentrate), and sodium pyruvate (100 mM) were purchased from Biochrom (Berlin, Germany).

### 4.2. Metabolomics Workflow

For the metabolomics study, the *E. coli* strains EcN, 83972 and CFT073 were each cultivated in triplicate. Since three different colonies from each strain were taken for the cultivation in different flasks, they were regarded as three biological replicates. The used culture medium was based on minimum essential medium (MEM), a precisely defined medium. In addition, the culture medium was treated and incubated without bacteria under the same conditions as well as used as a blank sample for the further metabolomics workflow during the next four steps.

#### 4.2.1. Cultivation of the E. coli Strains EcN, 83972, and CFT073

For the culture medium, 5 mL of L-alanyl-L-glutamine (200 mM), 5 mL non-essential amino acids (100x-concentrate) and 5 mL of sodium pyruvate (100 mM) were added to 1 L of MEM. Initially, overnight cultures of each *E. coli* strain (EcN, 83972, and CFT073) were prepared. For this purpose, three individual colonies of each strain were picked from a lysogeny broth (LB) agar plate and used as inoculum of three overnight cultures in MEM per strain. The cultures were grown statically overnight at 37 °C.

Each overnight culture was used to inoculate in a 1:100 ratio 10 mL of fresh MEM in 50-mL Erlenmeyer flasks. One flask was prepared with culture medium only to be used as a blank sample. The cultures were cultivated for 6 h at 37 °C without shaking. After 6 h of cultivation, a sample of each culture was taken for measurement of the OD value at 600 nm (Ultrospec 2100^®^, GE Healthcare, Freiburg, Germany). The remaining cultures were centrifuged (6300× *g*, 5 min at 4 °C, Centrifuge 5804 R, Eppendorf AG, Hamburg, Germany) and the supernatants were sterile filtrated with Fast Flow & Low Binding Millipore Express^®^ PES Membrane (Millex^®^-GP, pore size 0.22 µm, Merck, Darmstadt, Germany). The samples were stored at −20 °C until analyses. 

In addition to the present study, LB medium was also used under the same conditions. Here, Ybt could only be detected in very low concentrations in the metabolome of EcN and 83972. Additionally, different cultivation times and different initial pH values of the medium were investigated initially for the cultivation of EcN in MEM. Furthermore, the cultivation of statically grown cultures and cultures shaken during growth were compared (data not shown). Finally, the experiments were carried out as described above with batch cultures in MEM, because these culture conditions reproducibly resulted in the most favorable yields of Ybt.

#### 4.2.2. Sample Preparation, Chromatographic Conditions, and Data Acquisition by LC-HRMS Analysis

For sample preparation, the samples were defrosted and shaken on a vortex shaker. The analysis was carried out in triplicate for each sample by liquid chromatography with high-resolution mass spectrometric.

The chromatographic separation was performed using a Nexera XR LC system (Shimadzu, Duisburg, Germany). A Nucleodur^®^ C18 Gravity-SB column (150 × 2 mm, 3 µm) with a 4 × 2 mm Gravity SB guard column (Macherey-Nagel, Düren, Germany) was taken for separation and as the mobile phase, ACN and water (+0.1% FA each) were used for gradient elution. The further parameters for the chromatographic conditions are reported in the Appendix A.

For the data acquisition, an LTQ Orbitrap XL^TM^ mass spectrometer with heated electrospray ionization (HESI) (Thermo Fisher Scientific, Dreieich, Germany) in positive ionization mode was used. The further parameters are reported in the Appendix A. The data analysis was carried out using the Xcalibur^TM^ software (Thermo Fisher Scientific, Dreieich, Germany).

#### 4.2.3. Data Processing and Statistical Analysis

The raw data files of all measured samples of the three *E. coli* strains and the blank sample each measured in three technical replicates were loaded into the MZmine 2 software (version 2.33) [46] without previous conversion to determine features lists, which depends on the *m*/*z*, retention times and intensities of the peaks. The parameters for all steps used for the data processing can be found in the (Appendix A). After exporting the final feature list to a CSV file, the comparison of the features was carried out using MetaboAnalyst [63] for statistical analysis. After the export of the CSV data file to MetaboAnalyst, the major steps, such as including missing values and logarithmic transformation of data, were carried out and no normalization of data was applied. Using the multivariate and univariate statistical analysis for comparing samples, the PCA score plot, the corresponding PCA loadings plot, and the volcano plot were received.

#### 4.2.4. Identification and Characterization of Metabolites by LC-MS/HRMS

To obtain more structural data and for the prediction of molecular formulas, the isotope pattern and different fragmentation experiments (MS^2^, MS^3^, and MS^4^) were used. A mass list of the ions to be fragmented as well as the in-source fragmentation were used for this measurement. Compared to the data acquisition for the metabolomics study, differing parameters of the fragmentation experiments are reported in the Appendix A.

### 4.3. Comparison of the Occurrence of Ybt and Several Derivatives in the Metabolome of Different E. coli Strains

The development of the more sensitive liquid chromatography with the tandem mass spectrometry (LC-MS/MS) method, compared to the LC-HRMS method, was carried out with a purchased Ybt-standard (EMC microcollections GmbH, Tübingen, Germany). Since no reference standards were available for the various derivatives of Ybt, and presuming a similar ionization of these compounds, the various parameters of the potentials were adjusted on the basis of those optimized for Ybt. The different fragments for the use of the multiple reaction monitoring (MRM) mode were selected based on the fragmentation experiments using LC-MS/HRMS.

#### Sample Preparation, Chromatographic Conditions, and Data Acquisition by LC-MS/MS Analysis

The samples were defrosted and shaken on a vortex shaker. The analysis was carried out in triplicate for each sample by LC-MS/MS. For the chromatographic separation, an Agilent system, consisting of a 1260 series degasser (G4225A), 1260 series binary pump (G1312B), 1260 series column oven (G1316A), 1260 series automated liquid sampler (G1367E), 1260 series thermostat (G1330B), and 1260 series controller (G4208A), was used (Agilent Technologies, Ratingen, Germany). A Nucleodur^®^ C18 Gravity-SB column (100 × 2 mm, 3 µm) with a 4 × 2 mm Gravity SB guard column (Macherey-Nagel, Düren, Germany) was used as stationary phase and ACN and water (+0.1% FA each) for gradient elution. Additional parameters for the chromatographic conditions are reported in the Appendix A. Measurements were carried out on a QTRAP^®^ 5500 with electrospray ionization (ESI) (Sciex, Darmstadt, Germany) in MRM mode. The parameters of the ESI and the different transition of the metabolites are reported in the Appendix A. The data analysis was carried out using the Analyst 1.6.2 software (Sciex, Darmstadt, Germany).

### 4.4. Isolation of Ulbactin B from Bacterial Culture Supernatant of EcN

For the isolation of ulbactin B, the *E. coli* strain EcN was cultivated in the same culture medium (see Section 4.2.1) used for the metabolomics study. In total, 50 L of incubated medium were used for isolation. The cultivation and the pre-concentration of the culture supernatant used for the isolation was carried out in steps of 5 L.

#### 4.4.1. Cultivation of EcN for Isolation of Ulbactin B from the Culture Supernatant

For the overnight culture, one colony of *E. coli* strain EcN from an LB agar plate was transferred into 51 mL of culture medium in a 250-mL Erlenmeyer flask and incubated at 37 °C overnight without shaking. Afterwards, 1 mL of the culture was used for the measurement of the OD value at 600 nm and the remaining overnight culture was split into twice, 25 mL, which were added to 5-L flasks prepared with 2.5 L of culture medium each. These flasks were cultivated at 37 °C for 6 h without shaking.

After, the cultures were cooled down on ice and centrifuged (4780 × *g*, 20 min at 4 °C, Rotanta 460 RS, Hettich, Tuttlingen, Germany). The culture supernatant was sterile filtrated under vacuum into empty bottles using Nalgene™ Rapid-Flow™ Sterile Disposable Filter Units with PES Membrane (pore size 0.2 µm, Thermo Fisher Scientific, Dreieich, Germany). After filtration of half the culture, the remaining half was centrifuged a second time (6300 × *g*, 5 min at 4 °C, Centrifuge 5804 R, Eppendorf AG, Hamburg, Germany), receiving a more stable cell pellet to prevent the filter from clotting. The pellet was sterilized with ACN and discarded. The filtrated supernatant was stored at 4 °C overnight to use directly for the pre-concentration on the next day.

#### 4.4.2. Pre-concentration of 5 L Culture Supernatant

For pre-concentration, column material of 95 g Diaion^®^ HP-20 (Sigma-Aldrich, Steinheim, Germany) was prepared by incubation in MeOH for 15 min followed by incubation and equilibration in water for 15 min before use. Thereafter, a column was packed with the material and loaded with 5 L of culture supernatant. After washing the column with 150 mL of water, the elution was carried out using 620 mL of acetonitrile (ACN)+0.1% FA. The yellow eluate was collected in a round flask and the organic solvent removed. A total of 50 L of culture supernatant was pre-concentrated to 60 mL of eluate.

#### 4.4.3. Fractionated Solid Phase Extraction (SPE) of the Eluate of the Culture Supernatant 

The 60 mL of eluate were fractionated by the application of SPE to eliminate more matrix in preparation for concentrating the eluate further up. The fractionation was carried out by using Strata^®^ XL (10 g/60 mL, Phenomenex, Aschaffenburg, Germany). After conditioning with MeOH and equilibration with water, the SPE cartridge was loaded with 3–4 mL of the eluate in each case. The elution was carried out by different solvent mixtures consisting of ACN and water (+0.1% FA each). The steps of the fractionation are reported in Table 13.

Each fraction was, after evaporation and resolving in water, analyzed by LC-MS/MS. For the further isolation of ulbactin B, the fraction of the step number 6 (ACN/water/FA, 80/20/0.1, *v*/*v*/*v*) was used for preparative HPLC-UV after removing the organic solvent.

#### 4.4.4. Preparative HPLC-UV of the Fraction of Ulbactin B and Acquisition of NMR Data

Ulbactin B was finally isolated from the fraction after the application of SPE followed by use of the preparative HPLC-UV system (Varian, California, USA). For chromatographic separation, a Nucleodur^®^ Phenyl-Hexyl column (250 × 10 mm, 5 µm) with a 4 × 2 mm Phenyl-Hexyl guard column (Macherey-Nagel, Düren, Germany) was used as the stationary phase and a mobile phase of ACN and water (+0.1% FA each). The used gradient and the further parameters of the chromatographic separation are reported in the Appendix A. The detection was carried out by means of a diode array detector (DAD) at 255 nm. Ulbactin B showed a retention time of 15.7 min under these chromatographic conditions and was collected by peak. The organic solvent of the collected and united fractions of ulbactin B was removed. Afterwards, the solution of ulbactin B was purified by SPE using a Strata^®^ C18-E cartridge (500 mg/3 mL, Phenomenex, Aschaffenburg, Germany). After conditioning with MeOH and equilibration with water, the SPE cartridge was loaded with the solution of ulbactin B. After elution with MeOH (+ 0.1% FA), the solution was freeze-dried and 0.2 mg of ulbactin B (purity >90%) were obtained. These 0.2 mg of ulbactin B were redissolved in MeOD+TMSI and measured by 600 MHz NMR (Agilent Technologies, Ratingen, Germany). The data analysis was carried out using MestReNova 9.0 (Mestrelab Research S.L., Santiago de Compostela, Spain).

## 5. Conclusions

The metabolome comparison of the three closely related *E. coli* strains EcN, 83972, and CFT073 indicated that these isolates, despite their different phenotypes, shared under the conditions tested largely similar metabolomes. The fact that significantly different discriminative metabolomics features could be identified, which represent different and even so far unknown variants of the metallophore Ytb as well as the siderophore aerobactin, further supports the finding that metal acquisition is an important trait to maintain bacterial fitness and pathogenicity. In addition, the detection of multiple structurally modified Ybt derivatives corroborates that such molecules are not solely involved in metal uptake but could be expressed to serve other purposes, including bacterium–host interaction. It will be interesting to elucidate the molecular basis for the different structural modifications of the Ytb core structure, the underlying regulatory processes, and the biological function of the individual members of this metallophore family. LC-MS/HRMS-based metabolomic analyses will be very helpful in discovering the diversity of bacterial metabolomes and the impact of metallophores on bacterial fitness and pathogenicity.

## Figures and Tables

**Figure 1 metabolites-10-00221-f001:**
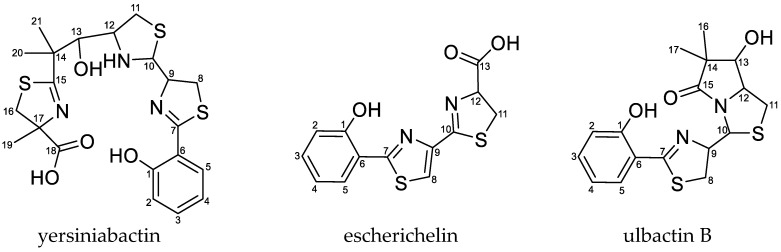
Structure of the siderophores yersiniabactin (Ybt) and escherichelin known for *Escherichia coli* (*E. coli*) and ulbactin B, which has not been described for *E. coli* before.

**Figure 2 metabolites-10-00221-f002:**
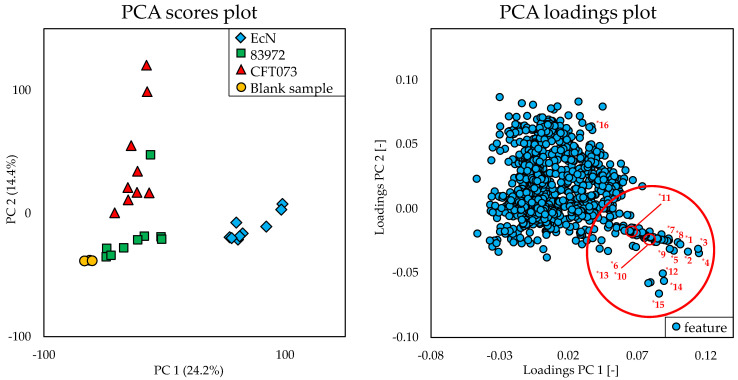
The principal component analysis (PCA) scores plot and the corresponding PCA loadings plot of the multivariate statistical analysis of the metabolome of the *Escherichia coli* (*E. coli*) strains EcN, 83972, and CFT073 cultivated in minimum essential medium (MEM) for 6 h and the blank sample treated under the same conditions. Three biological replicates of each strain were analyzed in triplicate. The labeled numbers of the red marked area correspond to the features of Table 1.

**Figure 3 metabolites-10-00221-f003:**
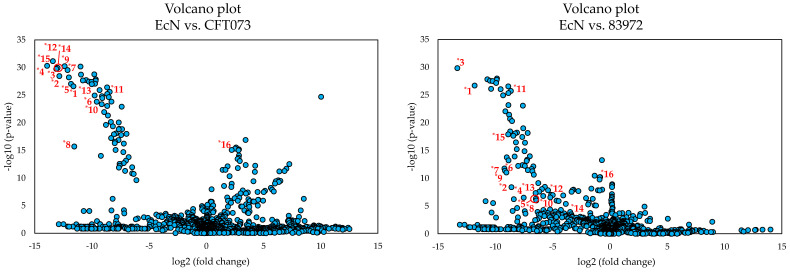
The volcano plots of the univariate statistical analysis of the metabolome of the *E. coli* strains EcN vs. CFT073 and EcN vs. 83972 cultivated in MEM for 6 h. Three biological replicates of each strain were analyzed in triplicate. The labeled numbers of the features correspond to the features of Table 1.

**Figure 4 metabolites-10-00221-f004:**
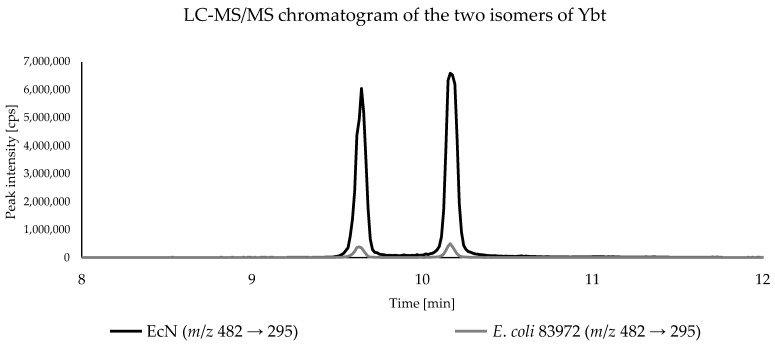
LC-MS/MS chromatogram of the culture supernatant of *E. coli* strains EcN and 83972, displaying the two isomers of Ybt with the mass spectrometric transition *m*/*z* 482→ 295.

**Figure 5 metabolites-10-00221-f005:**
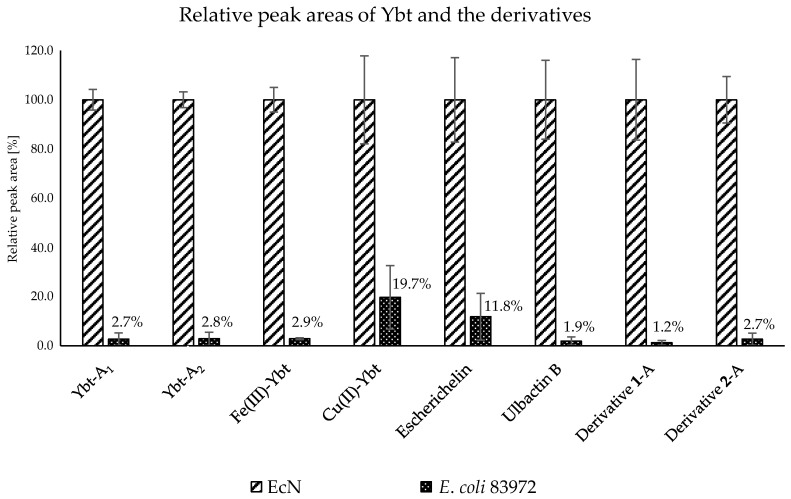
Relative peak areas of Ybt and the derivatives of the culture supernatant from *E. coli* strains EcN and 83972 after LC-MS/MS analysis.

**Figure 6 metabolites-10-00221-f006:**
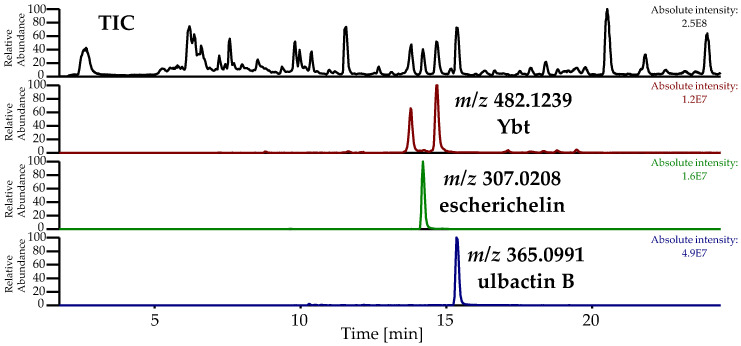
Total ion chromatogram (TIC) and extracted ion chromatogram (EIC) of pre-concentrated culture supernatant of EcN during the isolation of ulbactin B, displaying Ybt ([M+H]^+^: C_21_H_28_N_3_O_4_S_3_^+^, calculated: *m*/*z* 482.1236, measured: *m*/*z* 482.1239, Δm 0.5 ppm), escherichelin ([M+H]^+^: C_13_H_11_N_2_O_3_S_2_^+^, calculated: *m*/*z* 307.0206, measured: *m*/*z* 307.0208, Δm 0.8 ppm) and ulbactin B ([M+H]^+^: C_17_H_21_N_2_O_3_S_2_^+^, calculated: *m*/*z* 365.0988, measured: *m*/*z* 365.0991, Δm −0.8 ppm).

**Figure 7 metabolites-10-00221-f007:**
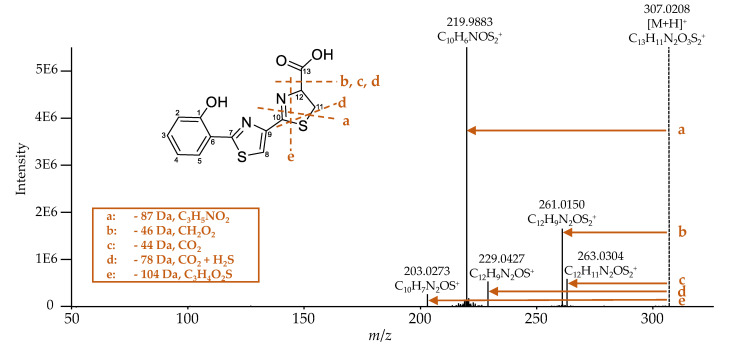
MS^2^ spectrum of escherichelin ([M+H]^+^: C_13_H_11_N_2_O_3_S_2_^+^, calculated: *m*/*z* 307.0206, measured: *m*/*z* 307.0208, Δm 0.8 ppm). It corresponds to that in literature [29], which also described the same NL of 87 (fragment a), 46 (fragment b), 78 (fragment d), and 104 Da (fragment e). Additionally, under this condition of the fragmentation experiment, the NL of 44 Da (fragment c) was detected as well.

**Figure 8 metabolites-10-00221-f008:**
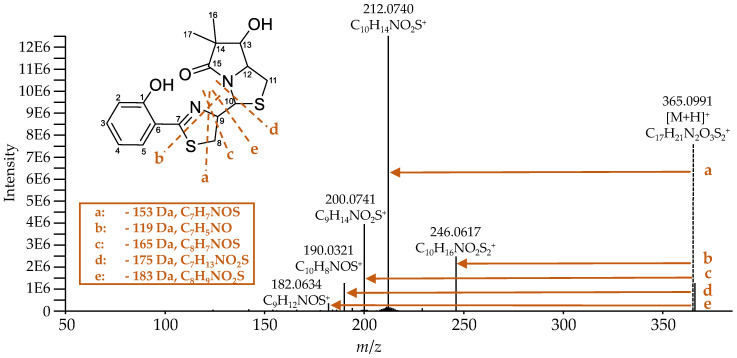
MS^2^ spectrum of ulbactin B ([M+H]^+^: C_17_H_21_N_2_O_3_S_2_^+^, calculated: *m*/*z* 365.0988, measured: *m*/*z* 365.0991, Δm −0.8 ppm). It shows an NL of 153 (fragment a), 119 (fragment b), 165 (fragment c), 175 (fragment d), and 183 Da (fragment e), which correspond to the proposed fragments.

**Figure 9 metabolites-10-00221-f009:**
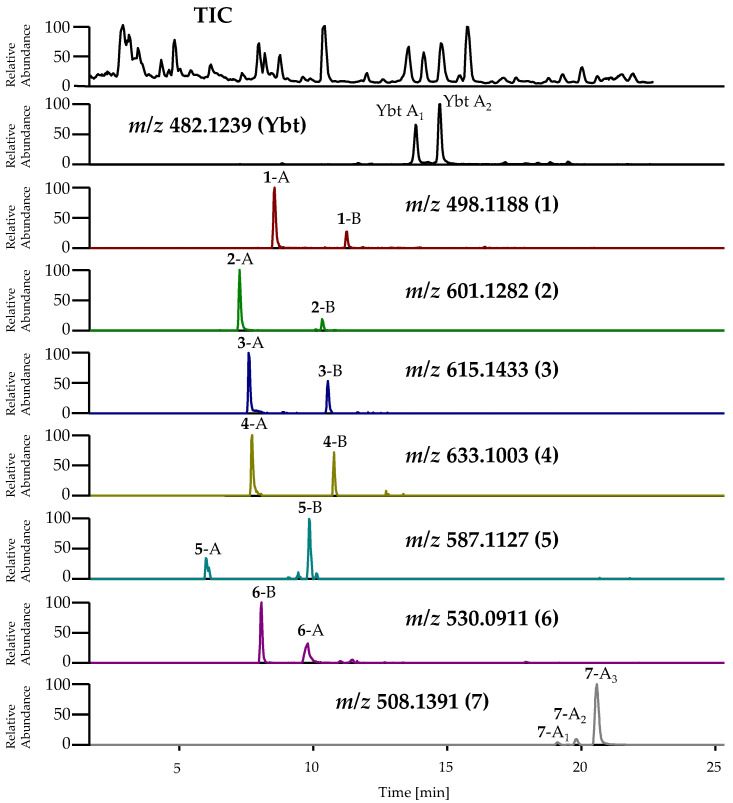
TIC and EIC of Ybt and its novel derivatives in the pre-concentrated supernatant of cultures of EcN from the isolation of ulbactin B. Analysis was carried out by LC-HRMS. The abbreviations for these metabolites are summarized in Appendix A. Ybt ([M+H]^+^: C_21_H_28_N_3_O_4_S_3_^+^, calculated: *m*/*z* 482.1236, measured: *m*/*z* 482.1239, Δm 0.5 ppm), **1** ([M+H]^+^: C_21_H_28_N_3_O_5_S_3_^+^, calculated: *m*/*z* 498.1186, measured: *m*/*z* 498.1188, Δm 0.5 ppm), **2** ([M+H]^+^: C_24_H_33_N_4_O_6_S_4_^+^, calculated: *m*/*z* 601.1277, measured: *m*/*z* 601.1282, Δm 0.8 ppm), **3** ([M+H]^+^: C_25_H_35_N_4_O_6_S_4_^+^, calculated: *m*/*z* 615.1434, measured: *m*/*z* 615.1433, Δm −0.2 ppm), **4** ([M+H]^+^: C_24_H_33_N_4_O_6_S_5_^+^, calculated: *m*/*z* 633.0998, measured: *m*/*z* 633.1003, Δm −0.8 ppm), **5** ([M+H]^+^: C_23_H_31_N_4_O_6_S_4_^+^, calculated: *m*/*z* 587.1121, measured: *m*/*z* 587.1127, Δm 1.0 ppm), **6** ([M+H]^+^: C_21_H_28_N_3_O_5_S_4_^+^, calculated: *m*/*z* 530.0906, measured: *m*/*z* 530.0911, Δm 0.9 ppm), **7** ([M+H]^+^: C_23_H_30_N_3_O_4_S_3_^+^, calculated: *m*/*z* 508.1393, measured: *m*/*z* 508.1391, Δm −0.4 ppm).

**Figure 10 metabolites-10-00221-f010:**
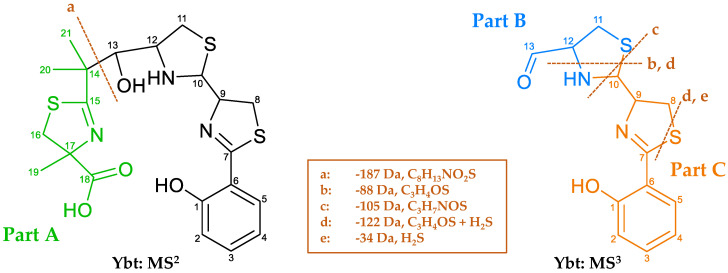
Fragmentation behavior of Ybt based on MS^2^ und MS^3^ experiments. Subdivision of the molecule into different parts: part A), including C14 to C21, part B) C11 to C13, and part C) C1 to C10.

**Figure 11 metabolites-10-00221-f011:**
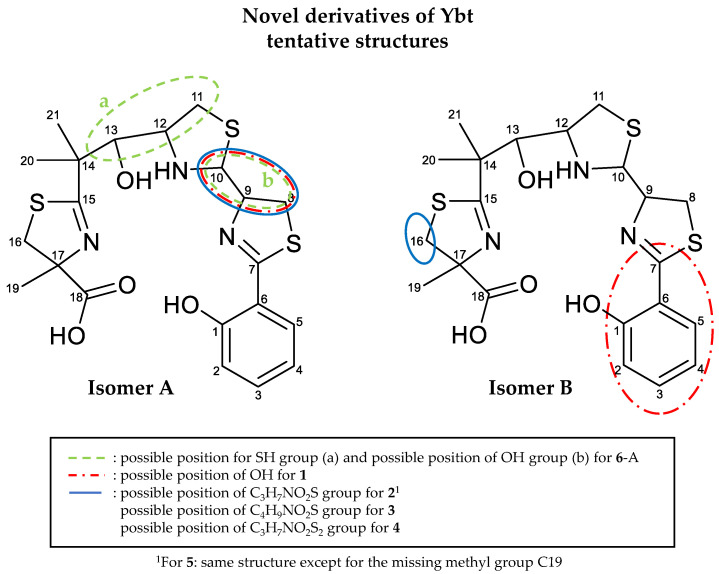
Proposed positions of the modifications of the novel derivatives of Ybt: **1** ([M+H]^+^: C_21_H_28_N_3_O_5_S_3_^+^, calculated: *m*/*z* 498.1186, measured: *m*/*z* 498.1188, Δm 0.5 ppm), **2** ([M+H]^+^: C_24_H_33_N_4_O_6_S_4_^+^, calculated: *m*/*z* 601.1277, measured: *m*/*z* 601.1282, Δm 0.8 ppm), **3** ([M+H]^+^: C_25_H_35_N_4_O_6_S_4_^+^, calculated: *m*/*z* 615.1434, measured: *m*/*z* 615.1433, Δm −0.2 ppm), **4** ([M+H]^+^: C_24_H_33_N_4_O_6_S_5_^+^, calculated: *m*/*z* 633.0998, measured: *m*/*z* 633.1003, Δm −0.8 ppm), **5** ([M+H]^+^: C_23_H_31_N_4_O_6_S_4_^+^, calculated: *m*/*z* 587.1121, measured: *m*/*z* 587.1127, Δm 1.0 ppm), **6** ([M+H]^+^: C_21_H_28_N_3_O_5_S_4_^+^, calculated: *m*/*z* 530.0906, measured: *m*/*z* 530.0911, Δm 0.9 ppm).

**Table 1 metabolites-10-00221-t001:** List of the features that represent the [M+H]^+^ ions for the most significant metabolites likely to be responsible for the significant differences of the metabolome of EcN compared to the metabolome of *E. coli* strains 83972 and CFT073 and the blank sample. Newly detected substances are labeled as bold numbers, isomeric forms showing differences in the retention time and fragmentation pattern are differentiated by capital letters (e.g., **1**-A and **1**-B), isomers differing in retention time but not in fragmentation pattern are differentiated with different subscript numbering (e.g., Ybt-A_1_ and Ybt-A_2_).

No. of Feature ^1^	*m*/*z* of [M+H]^+^	Retention Time [min]	Type of Ion	Predicted Molecular Formula of [M+H]^+^	Identification Based on MS/HRMS
*1	482.1229	13.25	[M+H]^+^	C_21_H_28_N_3_O_4_S_3_^+^ (Δm −1.5 ppm)	Ybt-A_1_ [18]
*2	295.0565		Fragment of Ybt-A_1_	C_13_H_15_N_2_O_2_S_2_^+^ (Δm −1.5 ppm)	
*3	482.1229	14.21	[M+H]^+^	C_21_H_28_N_3_O_4_S_3_^+^ (Δm −1.5 ppm)	Ybt-A_2_ [18]
*4	295.0565		Fragment of Ybt-A_2_	C_13_H_15_N_2_O_2_S_2_^+^ (Δm −1.5 ppm)	
*5	535.0340	10.99	[M+H]^+^	C_21_H_25_FeN_3_O_4_S_3_^+^ (Δm −2.1 ppm)	Fe(III)-Ybt [18]
*6	498.1178	8.32	[M+H]^+^	C_21_H_28_N_3_O_5_S_3_^+^ (Δm −1.5 ppm)	Derivative **1**-A
*7	311.0516		Fragment of **1**-A	C_13_H_15_N_2_O_3_S_2_^+^ (Δm −0.8 ppm)	
*8	601.1270	7.11	[M+H]^+^	C_24_H_33_N_4_O_6_S_4_^+^ (Δm −1.2 ppm)	Derivative **2**-A
*9	414.0606		Fragment of **2**-A	C_16_H_20_N_3_O_4_S_3_^+^ (Δm −1.1 ppm)	
*10	307.0201	13.88	[M+H]^+^	C_13_H_11_N_2_O_3_S_2_^+^ (Δm −1.5 ppm)	Escherichelin [29]
*11	365.0986	14.24	[M+H]^+^	C_17_H_21_N_2_O_3_S_2_^+^ (Δm −0.6 ppm)	Ulbactin B [31]
*12	323.0550	7.05	[M+H]^+^	C_11_H_19_N_2_O_3_S_3_^+^ (Δm −0.7 ppm)	**8**
*13	279.0651		Fragment of **8**	C_10_H_19_N_2_OS_3_^+^ (Δm −1.1 ppm)	
*14	543.0369	9.59	[M+H]^+^	C_21_H_26_CuN_3_O_4_S_3_^+^ (Δm −1.3 ppm)	Cu(II)-Ybt [20]
*15	355.9704		Fragment of Cu(II)-Ybt	C_13_H_13_CuN_2_O_2_S_2_^+^ (Δm 1.4 ppm)	
*16	565.2345	6.97	[M+H]^+^	C_22_H_37_N_4_O_13_^+^ (Δm −1.3 ppm)	Aerobactin [47]

^1^ The numbers correspond to the markings in the PCA loadings plot (Figure 2) and the volcano plots (Figure 3).

**Table 2 metabolites-10-00221-t002:** ^1^H and ^13^C NMR signals of the isolated ulbactin B in MeOD, which are in good agreement with data reported in the literature [31].

C	NMR Data (MeOD) from This Study		Configuration of Carbon Atom	NMR Data (CDCl_3_) from Literature [44]	
	δ_C_	δ_H_		δ_C_	δ_H_
1	160.3	-	C	159.1	-
2	117.8	6.95	CH	117.1	6.97
3	134.6	7.38	CH	133.5	7.35
4	120.2	6.91	CH	119.1	6.87
5	131.8	7.43	CH	130.8	7.40
6	117.5	-	C	116.1	-
7	175.2	-	C	174.4	-
8	34.5	3.34/3.51	CH_2_	33.9	3.23/3.42
9	82.1	5.07	CH	80.2	5.10
10	62.9	5.30	CH	63.0	5.37
11	30.3	2.99/3.29	CH_2_	30.0	3.00/3.31
12	67.6	4.60	CH	65.5	4.55
13	75.7	4.02	CH	76.0	4.05
14	51.4	-	C	49.8	-
15	179.9	-	C	177.3	-
16	18.8	1.12	CH_3_	18.1	1.19
17	24.2	1.25	CH_3_	23.7	1.3

**Table 3 metabolites-10-00221-t003:** Predicted molecular formulas of the novel derivatives based on mass spectrometric fragmentation experiments and the difference of these molecular formulas compared to Ybt.

*m*/*z* of the [M+H]^+^ Ion	Predicted Molecular Formula of M	Difference of the Predicted Molecular Formula to Ybt
482.1239 (Ybt)	C_21_H_27_N_3_O_4_S_3_	-
498.1188 (**1**)	C_21_H_27_N_3_O_5_S_3_	O
601.1282 (**2**)	C_24_H_32_N_4_O_6_S_4_	C_3_H_5_NO_2_S
615.1433 (**3**)	C_25_H_34_N_4_O_6_S_4_	C_4_H_7_NO_2_S
633.1003 (**4**)	C_24_H_32_N_4_O_6_S_5_	C_3_H_5_NO_2_S_2_
587.1127 (**5**)	C_23_H_30_N_4_O_6_S_4_	C_2_H_3_NO_2_S
530.0911 (**6**)	C_21_H_27_N_3_O_5_S_4_	OS
508.1391 (**7**)	C_23_H_29_N_3_O_4_S_3_	C_2_H_2_

**Table 4 metabolites-10-00221-t004:** *m*/*z* of the [M+H]^+^ ions, the typical NLs, and the resulting main fragments of Ybt and its novel derivatives derived from LC-MS/HRMS data. The most intense ion, in this case the fragment resulting from the typical neutral loss, was used for the MS^3^ experiments.

*m*/*z* of the [M+H]^+^ Ion	NL of 187 Da C_8_H_13_NO_2_S	NL of 173 Da C_7_H_11_NO_2_S	Measured Ion
482.1239 (Ybt-A_1_/Ybt-A_2_)	295.0570 ^1^ C_13_H_15_N_2_O_2_S_2_^+^	**-**	[M-187+H]^+^
498.1188 (**1**-A)	311.0520 ^1^ C_13_H_15_N_2_O_3_S_2_^+^	-	[M-187+H]^+^
601.1282 (**2**-A)	414.0610 ^1^ C_16_H_20_N_3_O_4_S_3_^+^	-	[M-187+H]^+^
615.1433 (**3**-A)	428.0763 ^1^ C_17_H_22_N_3_O_4_S_3_^+^	-	[M-187+H]^+^
633.1003 (**4**-A)	446.0329 ^1^ C_16_H_20_N_3_O_4_S_4_^+^	-	[M-187+H]^+^
587.1127 (**5**-A)	-	414.0610 ^1^ C_16_H_20_N_3_O_4_S_3_^+^	[M-173+H]^+^
530.0911 (**6**-A)	343.0241 ^1^ C_13_H_15_N_2_O_3_S_3_^+^	-	[M-187+H]^+^
508.1391 (**7**-A_1–3_)	321.0726 ^1^ C_15_H_17_N_2_O_2_S_2_^+^	-	[M-187+H]^+^

^1^ Used for MS^3^ experiments.

**Table 5 metabolites-10-00221-t005:** Selected NLs recorded during MS^3^ experiments of Ybt and its novel derivatives by application of LC-HRMS. All recorded fragmentation spectra are shown in the Appendix A. The resulting fragments [M-187-X+H]^+^ caused by the specific NL [X] listed on top of the columns are noted in the table. The most intense ion was used for MS^4^ experiments.

*m*/*z* of [M-187+H]^+^ (M)	Neutral Loss				
	18 Da H_2_O	34 Da H_2_S	88 Da C_3_H_4_OS	105 Da C_3_H_7_NOS	122 Da C_3_H_6_OS_2_
295.0570 (Ybt-A_1_/Ybt-A_2_)	277.0464 C_13_H_13_N_2_OS_2_^+^	261.0694 C_13_H_13_N_2_O_2_S^+^	207.0586 C_10_H_11_N_2_OS^+^	190.0320 C_10_H_8_NOS^+^	173.0709 C_10_H_9_N_2_O^+^
311.0520 (**1**-A)	293.0413 C_13_H_13_N_2_O_2_S_2_^+^	MS^4,2^	223.0533 C_10_H_11_N_2_O_2_S^+^ (MS^4^)	-	-
414.0610 ^1^ (**2**-A)	-	-	326.0629^3^ C_13_H_16_N_3_O_3_S_2_^+^ (+ MS^4,2^)	-	-
428.0763 (**3**-A)	-	MS^4,2^	MS^4,2^	-	MS^4,2^
446.0329 (**4**-A)	-	-	358.0348 C_13_H_16_N_3_O_3_S_3_^+^ (+ MS^4,2^)	-	-
343.0241 (**6**-A)	325.0135 C_13_H_13_N_2_O_2_S_3_^+^	-	-	-	-
321.0726 (**7**-A_1–3_)	-	-	-	-	-

^1^ Is also equivalent to the main fragment of (**5**-A) for the [M-173+H]^+^ ion; ^2^ detectable in MS^4^ experiments; ^3^ low intensity in the fragmentation spectra (Appendix A).

**Table 6 metabolites-10-00221-t006:** Selected fragments resulting from different NLs recorded during MS^3^ experiments of Ybt and its novel derivatives by application of LC-HRMS. All recorded fragmentation spectra are shown in the Appendix A. The specific NL [X] of a fragment leading to the ions [M-187-X+H]^+^ listed on top of the columns is noted in the table. The most intense ion was used for MS^4^ experiments.

*m*/*z* of [M-187+H]^+^ (M)	Fragment [M-187-X+H]^+^				
	173.0709 C_10_H_9_N_2_O^+^	190.0320 C_10_H_8_NOS^+^	261.0694 C_13_H_13_N_2_O_2_S^+^	293.0413 C_13_H_13_N_2_O_2_S_2_^+^	295.0576 C_13_H_15_N_2_O_2_S_2_^+^
295.0570 (Ybt-A_1_/Ybt-A_2_)	122 Da C_3_H_6_OS_2_	105 Da C_3_H_7_NOS	34 Da H_2_S	-	-
311.0520 (**1**-A)	-	-	-	18 Da H_2_0	-
414.0610 ^1^ (**2**-A)	241 Da C_6_H_11_NO_3_S_3_	-	153 Da C_3_H_7_NO_2_S_2_	121 Da C_3_H_7_NO_2_S	-
428.0763 (**3**-A)	255 Da C_7_H_13_NO_3_S_3_	-	167 Da C_4_H_9_NO_2_S_2_	135 Da C_4_H_9_NO_2_S	133 Da C_4_H_7_NO_2_S
446.0329 (**4**-A)	273 Da C_6_H_11_NO_3_S_4_	-	185 Da C_3_H_7_NO_2_S_3_	153 Da C_3_H_7_NO_2_S_2_	151 Da C_3_H_5_NO_2_S_2_
343.0241 (**6**-A) ^2^	-	-	-	-	-
321.0726 (**7**-A_1–3_)	-	131 Da C_5_H_9_NOS	60 Da C_2_H_4_S	-	-

^1^ Is also equivalent to the main fragment of (**5**-A) for the [M-173+H]^+^ ion; ^2^ fragments were not observed during MS^3^.

**Table 7 metabolites-10-00221-t007:** Selected fragments caused by different NLs recorded during MS^4^ experiments of Ybt and its novel derivatives by application of LC-HRMS. All recorded fragmentation spectra are shown in the Appendix A. The specific NLs [X] refer to the NLs shown in Table 5 and Table 6. These NLs combined with the specific NLs [Y] listed in this table lead to the fragment ions [M-187-X-Y+H]^+^ listed on top of the columns.

*m*/*z* of [M-187-X+H]^+^ (M)	Fragment [M-187-X-Y+H]^+^			
	120.0441 C_7_H_6_NO^+^	173.0709 C_10_H_9_N_2_O^+^	174.0547 C_10_H_8_NO_2_^+^	190.0320 C_10_H_8_NOS^+^
190.0319 (Ybt-A_1_/Ybt-A_2_)	70 Da C_3_H_2_S	-	-	-
208.0426 (**1**-A)	88 Da C_3_H_4_OS	-	34 Da H_2_S	18 Da H_2_O
293.0414 (**2**-A) ^2^	-	-	-	-
295.0570 (**3**-A) ^1^	-	122 Da C_3_H_6_OS_2_	-	-
293.0412 (**3**-A) ^1^	-	120 Da C_3_H_4_OS_2_	-	-
261.0690 (**4**-A)	-	88 Da C_3_H_4_OS	-	-
- -(**6**-A) ^2^	-	-	-	-
- -(**7**-A_1–3_) ^2^	-	-	-	-

^1^ 3-A showed two intense ions used for MS^4^; ^2^ fragments were not observed during MS^4^.

**Table 8 metabolites-10-00221-t008:** Selected fragments caused by different NLs recorded during MS^4^ experiments of Ybt and its novel derivatives by application of LC-HRMS. All recorded fragmentation spectra are shown in the Appendix A. The specific NLs [X] refer to the NLs shown in Table 5 and Table 6. These NLs combined with the specific NLs [Y] listed in this table lead to the fragment ions [M-187-X-Y+H]^+^ listed on top of the columns.

*m*/*z* of [M-187-X+H]^+^ (M)	Fragment [M-187-X-Y+H]^+^		
	205.0428 C_10_H_9_N_2_OS^+^	207.0583 C_10_H_11_N_2_OS^+^	261.0694 C_13_H_13_N_2_O_2_S^+^
190.0319 (Ybt-A_1_/Ybt-A_2_) ^2^	-	-	-
208.0426 (**1**-A) ^2^	-	-	-
293.0414 (**2**-A)	88 Da C_3_H_4_OS	-	-
295.0570 (**3**-A) ^1^	-	88 Da C_3_H_4_OS	34 Da H_2_S
293.0412 (3-A) ^1,2^	-	-	-
261.0690 (**4**-A) ^2^	-	-	-
- (**6**-A) ^2^	-	-	-
- (7-A_1–3_) ^2^	-	-	-

^1^ 3-A showed two intense ions used for MS^4^; ^2^ fragments were not observed during MS^4^.

**Table 9 metabolites-10-00221-t009:** Comparison of the most important NLs [X] and fragments [M-187-X+H]^+^ of the main fragment of the A-labelled isomer of the derivatives **2** and **4**.

*m*/*z* of [M-187+H]^+^ (M)	87 Da C_3_H_5_NO_2_	153 Da C_3_H_7_NO_2_S_2_	121 Da C_3_H_7_NO_2_S	119 Da C_3_H_5_NO_2_S
414.0610 (**2**-A)	327.0292 C_13_H_15_N_2_O_2_S_3_^+^	261.0692 C_13_H_13_N_2_O_2_S^+^	293.0410 C_13_H_13_N_2_O_2_S_2_^+^	-
446.0329 (**4**-A)	359.0015 C_13_H_15_N_2_O_2_S_4_^+^	293.0410 C_13_H_13_N_2_O_2_S_2_^+^	325.0135 C_13_H_13_N_2_O_2_S_3_^+^	327.0294 C_13_H_15_N_2_O_2_S_3_^+^

**Table 10 metabolites-10-00221-t010:** Selected main fragments caused by different NL recorded during MS^2^ experiments for the B isomers of the novel derivatives of Ybt by application of LC-MS/HRMS. All recorded fragmentation spectra are shown in the Appendix A. The specific NL [X] of a fragment leading to the ions [M-X+H]^+^ listed on top of the columns are noted in the table.

*m*/*z* of [M+H]^+^ Ion (M)	*m*/*z* of Main Fragment 365.0986 [M-X+H]^+^ ion C_17_H_21_N_2_O_3_S_2_^+^	*m*/*z* of Main Fragment 378.0975 [M-X+H]^+^ ion C_14_H_24_N_3_O_3_S_3_^+^	*m*/*z* of Main Fragment 466.1463 [M-X+H]^+^ ion C_21_H_28_N_3_O_5_S_2_^+^
498.1188 (**1**-B) C_21_H_28_N_3_O_5_S_3_^+^	-	120 Da C_7_H_4_O_2_	-
601.1282 (**2**-B) C_24_H_33_N_4_O_6_S_4_^+^	236 Da C_7_H_12_N_2_O_3_S_2_	-	-
615.1433 (**3**-B) C_25_H_35_N_4_O_6_S_4_^+^	250 Da C_8_H_14_N_2_O_3_S_2_	-	-
633.1003 (**4**-B) C_24_H_33_N_4_O_6_S_5_^+^	268 Da C_7_H_12_N_2_O_3_S_2_	-	-
587.1127 (**5**-B) C_23_H_31_N_4_O_6_S_4_^+^	222 Da C_6_H_10_N_2_O_3_S_2_	-	-
530.0911 (**6**-B) C_21_H_28_N_3_O_5_S_4_^+^	-	-	S_2_ 64 Da

**Table 11 metabolites-10-00221-t011:** Selected NLs recorded during MS^2^ experiments for the B isomers of the novel derivatives of Ybt by application of LC-MS/HRMS. All recorded fragmentation spectra are shown in the Appendix A. The resulting fragments [M-X+H]^+^ caused by the specific NL [X] listed on top of the columns are noted in the table.

*m*/*z* of [M+H]^+^ (M)	NEUTRAL LOSS			
	18 Da H_2_O	121 Da C_3_H_7_NO_2_S	135 Da C_4_H_9_NO_2_S	153 Da C_3_H_7_NO_2_S_2_
498.1188 (**1**-B)	480.1079 C_21_H_26_N_3_O_4_S_3_^+^	-	-	-
601.1282 (**2**-B)	583.1171 C_24_H_31_N_4_O_5_S_4_^+^	480.1072 C_21_H_26_N_3_O_4_S_3_^+^	-	-
615.1433 (**3**-B)	-	-	480.1064 C_21_H_26_N_3_O_4_S_3_^+^	-
633.1003 (**4**-B)	-	-	-	480.1072 C_21_H_26_N_3_O_4_S_3_^+^
587.1127 (**5**-B)	-	466.0921 C_20_H_24_N_3_O_4_S_3_^+^	-	434.1205 C_20_H_24_N_3_O_4_S_2_^+^

**Table 12 metabolites-10-00221-t012:** Selected NLs recorded during MS^2^ experiments for the B isomers of the novel derivatives of Ybt by application of LC-HRMS. All recorded fragmentation spectra are shown in the Appendix A. The resulting fragments [M-X+H]^+^ caused by the specific NL [X] listed on top of the columns are noted in the table (continued).

*m*/*z* of [M+H]^+^ (M)	Neutral Loss			
	187 Da C_8_H_13_NO_2_S	207 Da C_10_H_9_NO_2_S	310 Da C_13_H_14_N_2_O_3_S_2_	320 Da C_12_H_20_N_2_O_4_S_2_
498.1188 (**1**-B)	311.0514 C_13_H_15_N_2_O_3_S_2_^+^	291.0832 C_11_H_19_N_2_O_3_S_2_^+^	188.0740 C_8_H_14_NO_2_S^+^	-
601.1282 (**2**-B)	-	-	291.0830 C_11_H_19_N_2_O_3_S_2_^+^	-
615.1433 (**3**-B)	-	-	-	295.0563 C_13_H_15_N_2_O_2_S_2_^+^
633.1003 (**4**-B) ^1^	-	-	-	-
587.1127 (**5**-B) ^1^	-	-	-	-

^1^ NLs were not observed during MS^2^.

**Table 13 metabolites-10-00221-t013:** Steps of the fractionation of the bacterial eluate by the application of solid phase extraction (SPE).

Step of Fractionation	Volume [mL]	ACN/Water/FA (*v/v/v*)
1	120	15/85/0.1
2	60	20/80/0.1
3	60	25/75/0.1
4	60	30/70/0.1
5	60	35/65/0.1
6	80	80/20/0.1
7	80	100/0/0.1

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
