# Peer review of "Metabolomics Study on Pathogenic and Non-pathogenic E. coli with Closely Related Genomes with a Focus on Yersiniabactin and Its Known and Novel Derivatives"

_metabolites, 2020, doi:10.3390/metabo10060221_

Round 1

Reviewer 1 Report

The authors describe the discovery of apparent derivatives of a metalloprotein from closely related strains of E. coli. Metal scavenging proteins have significance in many virulence models, UTIs included. They found a versitile protein, Ybt, to be in complex with multiple metals but could not fully analyze all derivatives by NMR due to concentration issues. It is unfortunate that other media types or concentration schemes were not (apparently) tried to increase the concentration of these derivatives. Sample preparation is 3/4 of the battle in mass spectrometry. Did the authors consider other media types or growth phases? It is possible that some of these metal scavengers are secreted during different phases of the life cycle. This work is important. Increasingly it is becoming clear that genomics can not stand alone to describe everything observed in various pathogenic models. Metabolomics of this precise nature can help us understand the precise mechanisms employed by pathogens during infection and may provide data that can be used for more targeted therapies. Authors should talk about this more obviously in their introduction. 

Authors provide interesting comparisons of various chemical moieties of Ybt by comparing NLs of various fragments.  The resolution the authors claim with their mass spec experiments is impressive and their use of MS, MSMS, and MS3 data to suss out structural differences is impressive.

It would have been interesting to see them employ a different ionizatin technique to supplement some of their less rigorously supported findings. Cysteine bonds can be confirmed or analyzed by mass spec in many ways, which makes the "suspected" part of their side-chain analysis regrettable but doesn't detract from the paper too badly. 

Generally, there are sentence structure issues throughout the paper that need to be addressed by an English-writer. Examples are below, but the entire paper should be checked for sentence structure and word usage. 

Line 379: "Due to a low concentration of the other unknown derivatives, it is assumed that they could not be detected 380 during the metabolomics study."

Line 93: "These three strains have already been compared a lot on the genetic level due to the different 93 expression of the respective phenotypes."

 Suggest changing it to "The gentoypes of these three strains have been analyzed and compared extensively due to extensive phenotypic variation."

These three strains have already been compared a lot on the genetic level due to the different 93 expression of the respective phenotypes.

Line 105: The E. coli strains were cultivated for 6 h at 37 °C without shaking in minimum essential medium 105 (MEM). Additionally, culture medium without bacteria was treated under the same conditions as 106 blank sample. 

-- Move to materials and methods.

Figure 6. : What software made these TIC graphs? They look almost hand drawn...

Figure 9: Same comment.

In closing, this paper is impressive but also a bit overwhelming. I understand that the author's may have wanted to provide as much data as possible, but if there is a way to decide how much of this really needs to be in the main body versus supplemental, or what exposition can be trimmed, it would help readers. 

Author Response

see enclosed file

Reviewer 2 Report

Reviewer comments for Metabolites manuscript 810568

Schulz et al have presented their investigation into the exometabolome (or secreted metabolome) of three E.coli strains (EcN, 83972 and CFT073), which are closely related but phenotypically different – as demonstrated by their pathogenicity.  The researchers used multiple hyphenated-mass spectrometry techniques to determine whether there was a significant differences between the E.coli strains in the metabolites that were secreted, when grown under identical conditions.  The results of this study show that while there is a lot of consistency in the exometabolomes of the strains, there are also distinct metabolites that have been identified that discriminate between strains.  These findings also support the fact that metal acquisition is an important trait to maintain bacterial health and pathogenicity.  A number of interesting structural modifications and novel metabolites were putatively identified, with excellent supporting mass spectrometric analysis.

Overall, this study is very well done, with a thorough application and explanation of the mass spectrometric analysis.  I have very few concerns about this manuscript and would be happy for it to be published.  The minor concerns and revisions are documented below – and mainly relate to lack of information in the methods description.

General comments:

  • It would be better if the authors referred to the analysis of the E.coli samples as the exometabolome as opposed to the metabolome. When reading the manuscript, the metabolome of the samples is referred to, which confusingly comes across as being the intracellular metabolites.  To avoid confusion it would be better to state up front that it is the exometabolome(or secreted metabolome).

Specific comments:

  • It’s not a deal breaker, but experimental design would have been much stronger if the authors had used more biological replicates rather than doing technical replicates of each biological replicate. All this does is test the reproducibility of the sample preparation and instrument analysis.  It doesn’t contribute to biological reproducibility.

  • The methods section requires more detail:
    • There is very limited description of the sample preparation. It’s stated that the centrifuged supernatant media was syringe filtered to sterilise, then frozen.  Supernatants were then defrosted and shaken on a vortex mixer.   Is this all that was done?  Was there a solvent extraction of the media supernatants?  Were internal standards added?
    • Was storage at -20degC advisable? It has been show in multiple studies that -20degC is not good enough to prevent metabolite breakdown.
    • There is no model information given for the Agilent LC system used for the LC-MS/MS analysis
    • For statistical analysis, the data was log transformed and normalised. What normalisation method was used?
    • Was the OD observed for each strain used for normalisation? And how did the formation of biofilm contribute to metabolic changes?

Author Response

see enclosed file
